# Interrogation of enhancer function by enhancer-targeting CRISPR epigenetic editing

Kailong Li[1,2,4], Yuxuan Liu[1,2,4], Hui Cao[1,2], Yuannyu Zhang[1,2], Zhimin Gu[1,2], Xin Liu[1,2], Andy Yu[1,3], Pranita Kaphle[1,2], Kathryn E. Dickerson[1,2], Min Ni[1,2] & Jian Xu [1,2]*

Tissue-specific gene expression requires coordinated control of gene-proximal and -distal *cis*-regulatory elements (CREs), yet functional analysis of gene-distal CREs such as enhancers remains challenging. Here we describe CRISPR/dCas9-based enhancer-targeting epigenetic editing systems, enCRISPRa and enCRISPRi, for efficient analysis of enhancer function in situ and in vivo. Using dual effectors capable of re-writing enhancer-associated chromatin modifications, we show that enCRISPRa and enCRISPRi modulate gene transcription by remodeling local epigenetic landscapes at sgRNA-targeted enhancers and associated genes. Comparing with existing methods, the improved systems display more robust perturbations of enhancer activity and gene transcription with minimal off-targets. Allele-specific targeting of enCRISPRa to oncogenic *TAL1* super-enhancer modulates *TAL1* expression and cancer progression in xenotransplants. Single or multi-loci perturbations of lineage-specific enhancers using an enCRISPRi knock-in mouse establish in vivo evidence for lineage-restricted essentiality of developmental enhancers during hematopoiesis. Hence, enhancer-targeting CRISPR epigenetic editing provides opportunities for interrogating enhancer function in native biological contexts.

[1] Children's Medical Center Research Institute, University of Texas Southwestern Medical Center, Dallas, TX 75390, USA. [2] Department of Pediatrics, Harold C. Simmons Comprehensive Cancer Center, and Hamon Center for Regenerative Science and Medicine, University of Texas Southwestern Medical Center, Dallas, TX 75390, USA. [3] SURF-Stem Cell Program, University of Texas Southwestern Medical Center, Dallas, TX 75390, USA. [4]These authors contributed equally: Kailong Li, Yuxuan Liu. *email: jian.xu@utsouthwestern.edu

Mammalian gene expression requires precisely regulated gene-proximal promoters and gene-distal *cis*-regulatory elements (CREs) such as transcriptional enhancers. Systematic annotation of human epigenomes has identified millions of putative CREs using correlative features such as chromatin accessibility and histone modifications[1–3]; however, the analysis of in vivo functions of these elements within their native chromatin remains difficult. This is in part because existing technologies often measure enhancer activity in heterologous assays without native chromatin, and because findings from these assays have not been causally connected with specific target genes or cellular functions during development.

Enhancers are *cis*-regulatory DNA sequences that are bound and regulated by transcription factors (TFs) and chromatin regulators in a highly cell-type and temporal-specific manner. Putative enhancers are operationally identified using epigenetic signatures including chromatin accessibility (DNase I hypersensitivity or ATAC-seq) and histone marks (H3K4me1/2 and H3K27ac)[4–6]. Unlike gene-proximal promoters, enhancers can regulate gene transcription over long distances in an orientation-independent and cell-type-specific manner[7]. As such, major challenges have limited the application of existing technologies in functional analysis of a specific enhancer in a mammalian genome. Reporter assays have historically been used to examine enhancer activity in heterologous cell models[7]. When combined with high-throughput sequencing, massively parallel reporter assays allow for quantitative analysis of the transcriptional activity of thousands of enhancers in particular cell types[8,9]. Together with transgenics, in vivo enhancer reporter assays enable evaluation of enhancer function during mammalian development[10]. These are powerful approaches for assaying TF-mediated transcriptional activity at enhancer DNA sequences, but they have some important limitations including the lack of local chromatin contexts and epigenetic features in heterologous assays, the often use of a general promoter such as SV40 rather than the enhancer's endogenous promoter, the inability to identify the target genes of enhancers, and the inadequacy to model combinatorial regulation by multiple enhancers at native chromatin. Additionally, conventional gene targeting or genome editing approaches have been utilized to knockout (KO) or mutate specific enhancers in cell lines or animal models[11,12]; however, they require genetic engineering that remains low-throughput and laborious. Furthermore, high-resolution saturating mutagenesis of *cis*-regulatory elements relies on loss-of-function and does not permit gain-of-function analyses[13,14].

Recently, major advances have been made in the modulation of endogenous gene expression by repurposing the CRISPR/Cas9 system[15–27]. By coupling the deactivated Cas9 (dCas9) to various activator (e.g. VP64[16–18] and p300[20,25]) or repressor (e.g. KRAB[16,17,21,25], LSD1[26], and DNMT3A/3L[23]) domains, transcriptional perturbations of specific genes were achieved. While the first-generation dCas9 activator or repressor complexes such as dCas9-VP64 and dCas9-KRAB can effectively modulate transcription when tethered to gene-proximal promoters, the effect declines rapidly when its target region moves away from proximal promoter sequences[17,19]. The development of second-generation dCas9 activators using combinations of effector proteins by dCas9 fusions and/or MS2-MCP scaffolding, including the synergistic activation mediator (SAM) system[19], SunTag[28] and VP64-p65AD-Rta (VPR)[29], improved the potency in targeted gene regulation; however, the strategies involving enhancer-targeting epigenetic modifiers are less characterized and the in vivo efficacy of dCas9-based epigenetic editing in development remains underexplored[30,31].

Here, we describe the enhancer-targeting CRISPR epigenetic editing systems, enCRISPRa and enCRISPRi, to interrogate enhancer function using dCas9 with dual effectors that specifically modulate epigenetic modifications at enhancers. Using the human β-globin locus control region (LCR), oncogenic *TAL1* super-enhancer (SE), and hematopoietic lineage-specific enhancers as examples, we show that enCRISPRa and enCRISPRi effectively modulate enhancer function in vitro, in xenografts and in vivo. Enhancer-targeting CRISPR epigenetic editing leads to locus-wide epigenetic reprogramming and interference with TF binding. Single- or multiloci in vivo enhancer perturbations using an enCRISPRi mouse model reveal lineage-specific requirements of developmental enhancers during hematopoietic lineage differentiation. Hence, the enhancer-targeting CRISPR epigenetic editing systems provide opportunities for functional interrogation of enhancers and other CREs in development and disease.

## Results

**Development of an enCRISPRa system for enhancer activation.** To assess the functional role of gene-distal enhancers, we devised the dCas9-based enhancer-targeting epigenetic perturbation systems for targeted modulation of enhancer activity in situ. Specifically, we employed the structure-guided sgRNA design by adding two MS2 hairpins, which is recognized by the MCP RNA-binding proteins[19]. For enhancer activation (enCRISPRa; Fig. 1a and Supplementary Fig. 1a), we fused dCas9 with the core domain of histone acetyltransferase p300, which catalyzes H3-Lys27 acetylation (H3K27ac)[32], together with the MS2-sgRNA sequence to recruit the MCP-VP64 activator domains. We further engineered the second version of enCRISPRa using dCas9-VP64 + MCP-p300 combination (Fig. 1a). Since H3K27ac is the hallmark of active enhancers[33], by doxycycline (Dox)-inducible expression of dCas9-p300 (or dCas9-VP64), sgRNA-MS2 and MCP-VP64 (or MCP-p300), we engineered the enhancer-targeting dual-activator systems.

As a proof-of-principle test of enCRISPRa in modulating CRE activity, we targeted enCRISPRa or single-effector dCas9 activators to a known *MYOD* enhancer[20] using sequence-specific sgRNAs in HEK293T cells (Fig. 1b). Compared to nontransduced control cells, dCas9 alone or dCas9-VP64 had no or minimal effect on *MYOD* expression, whereas dCas9-p300 significantly activated *MYOD* expression by 17.7-fold ($P < 0.001$ relative to control by a one-way ANOVA; Fig. 1b). Importantly, enCRISPRa with dCas9-VP64 + MCP-p300 (VP) or dCas9-p300 + MCP-VP64 (PV) combination led to enhanced *MYOD* activation (26.5- and 32.8-fold, $P < 0.001$ by a one-way ANOVA), indicating that the dual-effector enCRISPRa results in a more potent transcriptional activation of the targeted enhancer. We then focused on the well-established HS2 enhancer at the human β-globin locus[34] (Fig. 1c). The β-globin locus contains five β-like globin genes (*HBE1*, *HBG1*, *HBG2*, *HBD* and *HBB*) that are developmentally regulated by a shared upstream enhancer cluster or locus control region (LCR)[35]. The β-globin LCR consists of five discrete DNase I hypersensitive sites (HS1–HS5) in which HS2 functions as an erythroid-specific enhancer in transgenic assays[35], and provides a paradigm for studying tissue-specific and developmentally regulated gene transcription. Similar to the *MYOD* enhancer, the targeting of enCRISPRa to HS2 enhancer led to a more potent activation of β-like globin genes (*HBE1*, *HBG1/2* and *HBB*) compared to single-effector dCas9 activators (Fig. 1c). Given that two enCRISPRa versions (VP and PV) were largely comparable in gene activation (Fig. 1a–c), we focused on enCRISPRa with dCas9-p300 + MCP-VP64 in subsequent studies.

We next compared enCRISPRa with existing next-generation dCas9 activation methods such as dxCas9-VPR[22], SunTag[17] and SAM[19] (Fig. 1d, e). Notably, enCRISPRa displayed significantly

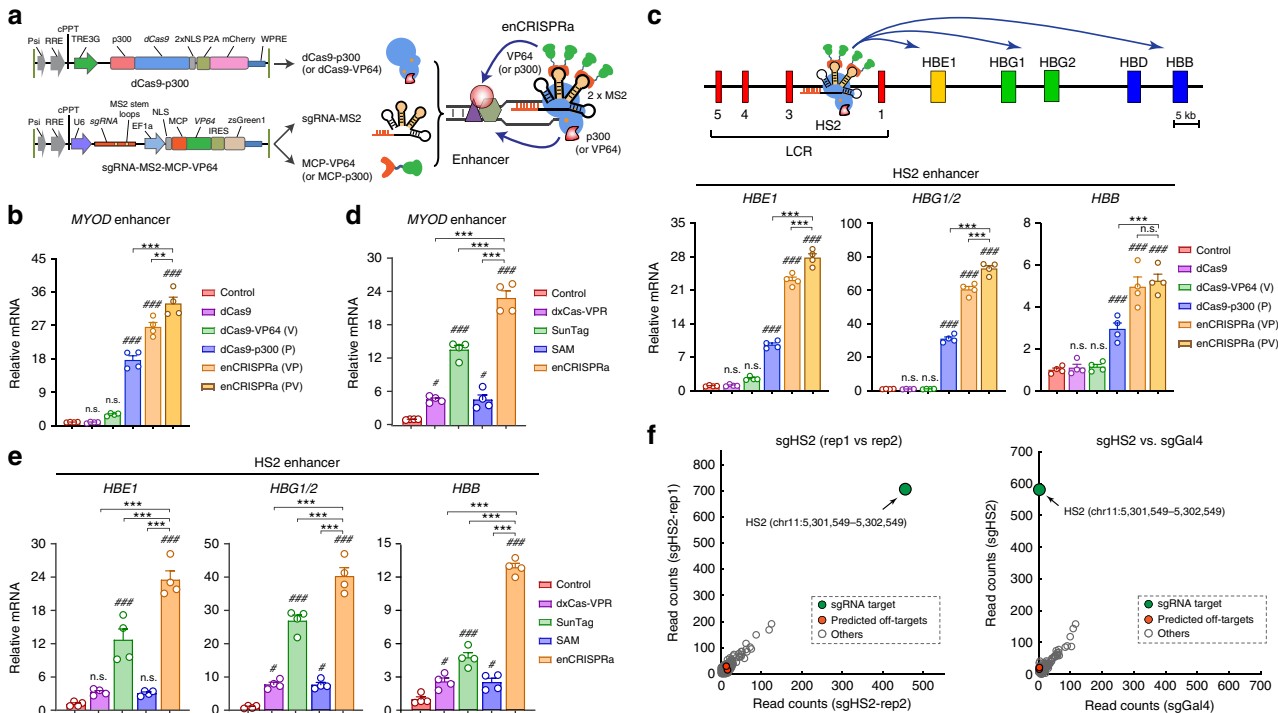

**Fig. 1 Development of the dual-activator enCRISPRa system. a** Schematic of enCRISPRa containing three components: a dCas9-p300 fusion protein, the sgRNA with two MS2 hairpins, and the MCP-VP64 fusion protein. **b** Expression of *MYOD* upon dCas9 alone, dCas9-VP64 (V), dCas9-p300 (P), dCas9-VP64 + MCP-p300 (enCRISPRa-VP) or dCas9-p300 + MCP-VP64 (enCRISPRa-PV)-mediated enhancer activation in HEK293T cells. mRNA expression relative to nontransduced cells (control) is shown as mean ± SEM (n = 4 experiments). The differences between control and dCas9 activators were analyzed by a one-way ANOVA. #P < 0.05, ###P < 0.001. The difference between different dCas9 activators were analyzed by a one-way ANOVA. *P < 0.05, **P < 0.01, ***P < 0.001, n.s. not significant. **c** Expression of *HBE1*, *HBG1/2* and *HBB* upon dCas9 alone, dCas9-VP64 (V), dCas9-p300 (P), or enCRISPRa (VP and PV)-mediated activation of the HS2 enhancer in HEK293T cells. mRNA expression relative to nontransduced cells is shown as mean ± SEM (n = 4 experiments) and analyzed by a one-way ANOVA. **d** Expression of *MYOD* upon dxCas9-VPR, SunTag, SAM or enCRISPRa-mediated enhancer activation in HEK293T cells. mRNA expression relative to nontransduced cells is shown as mean ± SEM (n = 4 experiments). The differences between control and dCas9 activators were analyzed by a one-way ANOVA. #P < 0.05, ###P < 0.001. The difference between different dCas9 activators were analyzed by a one-way ANOVA. ***P < 0.001. **e** Expression of *HBE1*, *HBG1/2* and *HBB* upon dxCas9-VPR, SunTag, SAM or enCRISPRa-mediated activation of HS2 in HEK293T cells. mRNA expression relative to nontransduced cells is shown as mean ± SEM (n = 4 experiments) and analyzed by a one-way ANOVA. **f** Genome-wide analysis of dCas9 binding in HEK293T cells expressing HS2-specific sgRNA (two replicates sgHS2-rep1 and sgHS2-rep2) or nontargeting sgGal4. Data points for the sgRNA target regions and the predicted off-targets are shown as green and red, respectively. The x- and y axis denote the normalized read counts (left) or mean normalized read counts from n = 2 experiments (right). Source data are provided as a Source Data file.

more robust activation of gene transcription compared to other dCas9 activators when targeted to the *MYOD* enhancer (Fig. 1d). At the β-globin locus, enCRISPRa-mediated HS2 enhancer activation led to 23.6-, 40.6- and 13.0-fold increases in expression of β-globin genes *HBE1*, *HBG1/2* and *HBB* relative to the nontransduced controls (P < 0.001 by a one-way ANOVA; Fig. 1e), respectively, which were significantly higher than other dCas9 activation methods. enCRISPRa showed comparable potency on gene activation when targeted to the *IL1RN* and *OCT4* promoters[20] (Supplementary Fig. 1b). Moreover, by examining dCas9 alone or dCas9-p300 in the presence or absence of MCP-VP64, together with sgRNAs with or without MS2 hairpins, we observed that the combination of dCas9-p300, MCP-VP64 and sgRNA with MS2 hairpins consistently enhanced gene activation compared to other combinations (Supplementary Fig. 1c, d), indicating that the assembled enCRISPRa complexes containing dual effectors (p300 and VP64) by MS2-MCP scaffolding are more effective in enhancer activation than individual effectors.

Although different dCas9 single- or multi-effector activators were noted to display variable potencies in gene activation in previous studies[19,36–38], the observed differences could be affected

by different cellular contexts, particular target genes, position of sgRNAs (e.g. promoter vs. enhancer), transfection conditions, and time to analyze gene expression[24]. Thus, it is important to note that our analyses were performed in the same cell lines using the same sgRNA sequences and transfection protocol, which enabled us to compare the efficacy of different dCas9 activators on enhancer perturbation side-by-side. Moreover, the analysis of dCas9 chromatin occupancy in cells coexpressing enCRISPRa and HS2-specific sgRNA (sgHS2) revealed highly reproducible binding at the targeted HS2 enhancer by independent ChIP-seq experiments (Fig. 1f; Supplementary Table 1). By comparing dCas9 binding in cells expressing sgHS2 or nontargeting sgRNA (sgGal4), we observed highly specific enrichment of dCas9 at the targeted HS2 enhancer with few potential off-target binding at the genome scale (Fig. 1f). No significant change in the expression of the nearest neighbor genes associated with the top potential off-targets was noted (Supplementary Fig. 1e–f).

**Development of an enCRISPRi system for enhancer repression.** We next devised a dual-effector epigenetic editing system for targeted enhancer inhibition (enCRISPRi-LK; Fig. 2a).

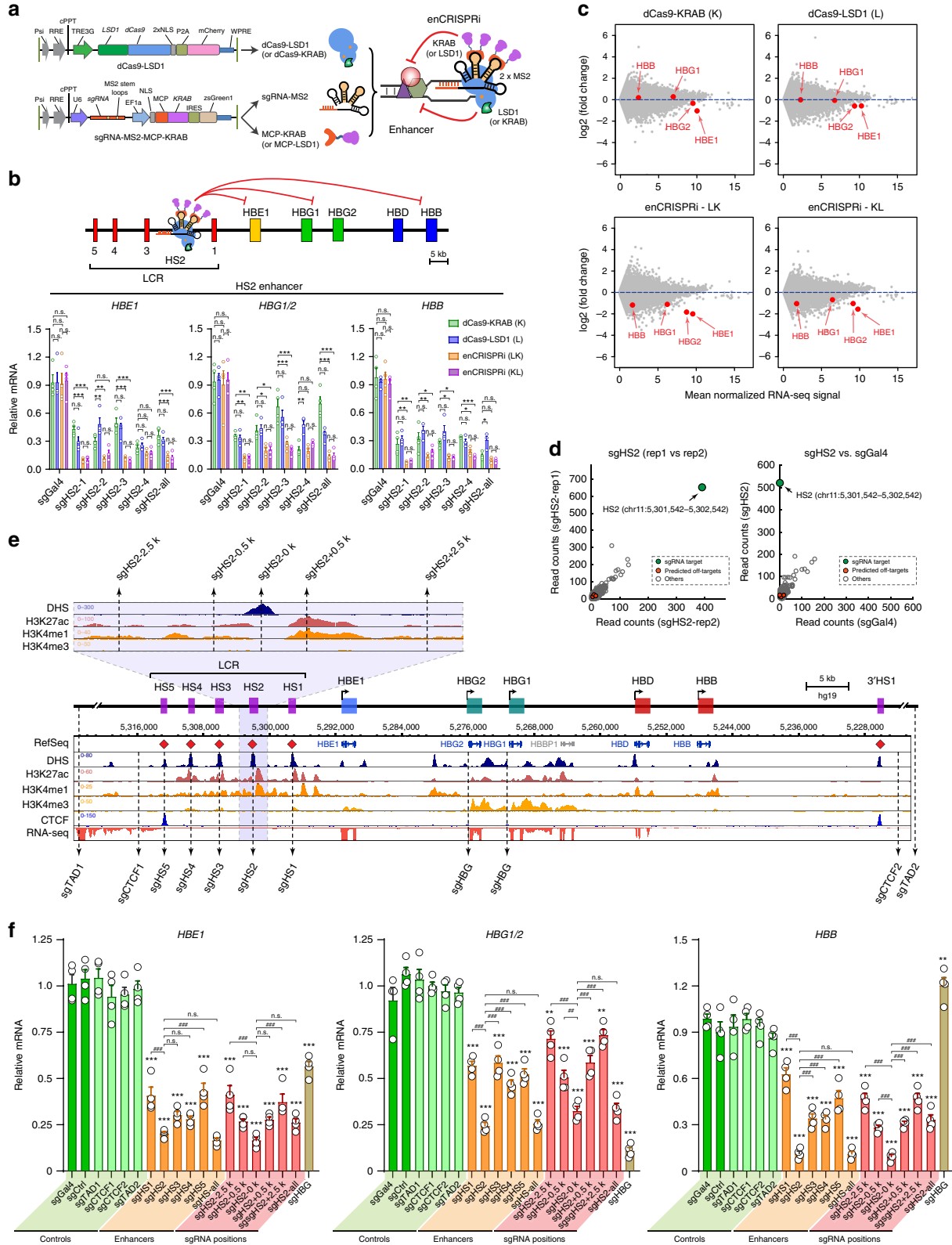

Specifically, we fused dCas9 with the lysine-specific demethylase LSD1 (or KDM1A), which catalyzes the removal of enhancer-associated H3-Lys4 mono- and di-methylation (H3K4me1/2)[39], together with MS2-sgRNA to recruit the MCP-KRAB repressor domains. As a proof-of-principle test, we targeted enCRISPRi to the HS2 enhancer in K562 erythroleukemia cells that highly express β-globin genes[34]. We achieved 3.4- to 13.7-fold repression

of β-globin genes (HBE1, HBG1/2 and HBB) by four independent sgRNAs (sgHS2-1 to sgHS2-4) relative to nontargeting sgGal4 (Fig. 2b). We further engineered the second version of enCRISPRi using dCas9-KRAB + MCP-LSD1 (enCRISPRi-KL) combination (Fig. 2a; Supplementary Fig. 1a). Targeting of either enCRISPRi complex to the HS2 enhancer by four independent sgRNAs achieved comparable and significant repression of β-globin genes

**Fig. 2 Development of the dual-repressor enCRISPRi system. a** Schematic of enCRISPRi containing a dCas9-LSD1 fusion protein, the sgRNA with two MS2 hairpins, and the MCP-KRAB fusion protein. **b** Expression of β-globin genes in K562 cells upon dCas9-KRAB (K), dCas9-LSD1 (L) or enCRISPRi (LK and KL)-mediated repression of the HS2 enhancer using four HS2-targeting sgRNAs individually (sgHS2-1 to sgHS2-4) or combined (sgHS2-all). The nontargeting sgGal4 was analyzed as the control. mRNA expression relative to nontransduced cells is shown as mean ± SEM ($n = 4$ experiments) and analyzed by a two-way ANOVA. *$P < 0.05$, **$P < 0.01$, ***$P < 0.001$, n.s. not significant. **c** RNA-seq profiles in K562 cells upon dCas9-KRAB, dCas9-LSD1 or enCRISPRi-mediated repression of the HS2 enhancer using four sgRNAs (sgHS2-all). Scatter plot is shown for each gene by the mean of log2 normalized RNA-seq signals as transcripts per million or TPM ($n = 2$ experiments) (x axis) and log2 fold changes of mean TPM in cells expressing sgHS2 and nontransduced cells (y axis). β-globin genes are indicated by red arrowheads. **d** Genome-wide analysis of dCas9 binding in K562 cells expressing HS2-specific sgRNA (sgHS2-rep1 and sgHS2-rep2) or nontargeting sgGal4. Data points for the sgRNA target regions and the predicted off-targets are shown as green and red, respectively. **e** Density maps are shown for DHS, ChIP-seq of H3K27ac, H3K4me1, H3K4me2, CTCF, and RNA-seq at the β-globin cluster (chr11: 5,222,500–5,323,700; hg19). The zoom-in view of the HS2 proximity region is shown on the top. Dashed lines denote the positions of sgRNAs. **f** Expression of β-globin genes in K562 cells coexpressing enCRISPRi and target-specific sgRNAs at various positions within the β-globin cluster, control sgRNAs (sgCtrl, sgTAD1, sgTAD2, sgCTCF1 and sgCTCF2) or nontargeting sgGal4. mRNA expression relative to nontransduced cells is shown as mean ± SEM. The differences between control sgGal4 and other sgRNAs were analyzed by a one-way ANOVA. *$P < 0.05$, **$P < 0.01$, ***$P < 0.001$, n.s. not significant. The differences between sgHS2 and other sgRNAs were analyzed by a one-way ANOVA. ##$P < 0.01$, ###$P < 0.001$. Source data are provided as a Source Data file.

(Fig. 2b). Comparing with the single-effector dCas9-KRAB (K) or dCas9-LSD1 (L) complex, the dual-repressor-containing enCRISPRi displayed stronger gene repression (Fig. 2b, c). In addition, enCRISPRi resulted in minimal changes (except β-globin genes) in global transcriptomics by RNA-seq (Fig. 2c). Moreover, ChIP-seq analyses revealed significant enrichment of dCas9 binding at the targeted HS2 enhancer with few off-targets by independent replicate experiments and/or by comparing to the nontargeting sgGal4 (Fig. 2d). No significant change in expression of the nearest neighbor genes associated with the top off-targets was observed (Supplementary Fig. 1g–h), indicating the locus-specific modulation of target gene transcription.

We further explored the effectiveness of enCRISPRi by targeting to single or multiple enhancers at the β-globin LCR (Fig. 2e). We focused on the enCRISPRi (LK) version given similar effects on gene repression when targeting enCRISPRi LK and KL to the HS2 enhancer (Fig. 2a–c). To this end, we designed sgRNAs for individual LCR enhancers (sgHS1 to sgHS5) and *HBG1/2* promoters (sgHBG). The β-globin locus is flanked by two CTCF-associated insulator elements at HS5 and 3′HS1[35]. As important negative controls, we designed sgRNAs targeting DNA sequences 2–4 kb outside of the β-globin insulators (sgCTCF1 and sgCTCF2), outside of the β-globin locus-containing topologically associating domain (TAD) (sgTAD1 and sgTAD2), or at a different chromosome (sgCtrl; chr2:211,337,408–211,337,427) (Fig. 2e; Supplementary Table 2). Upon stable coexpression of individual target-specific or control sgRNAs with enCRISPRi in K562 cells, we observed that sgHS2 resulted in more significant repression of all β-globin genes *HBE1*, *HBG1/2* and *HBB* (4.7- to 11.8-fold, $P < 0.001$ relative to sgGal4 by a one-way ANOVA) compared to other LCR enhancers (Fig. 2f), consistent with the prominent role of the HS2 enhancer for LCR function[34,35]. Further, coexpression of all five sgRNAs targeting LCR enhancers (sgHS-all) did not further repress β-globin genes (Fig. 2e, f). Notably, none of the control sgRNAs including two sgRNAs flanking β-globin insulators (sgCTCF1 and sgCTCF2) affected β-globin expression (Fig. 2e, f).

When targeted to gene-proximal promoters, dCas9-based epigenetic modulation may block TF binding and/or interfere with the formation of transcription initiation or elongation complexes[15–17]. By contrast, enhancer repression requires the interference with the function of specific enhancer-regulating TFs, chromatin regulators and their combinatorial activities. We reasoned that targeting enCRISPRi to the proximity of enhancer center may achieve maximal effects compared to enhancer distal sequences. To this end, we compared the efficacies of enCRISPRi by designing sgRNAs targeting the DNA sequences at the DNase

I hypersensitivity (DHS) peak summit at HS2 (sgHS2), or the sequences located 0.5 or 2.5 kb upstream or downstream of the HS2 enhancer (sgHS2-2.5k, sgHS2-0.5k, sgHS2 + 0.5k and sgHS2 + 2.5k; Fig. 2e), respectively. We achieved the most significant gene repression when targeting enCRISPRi to enhancer DHS peak summit (3.6- to 9.1-fold, $P < 0.001$ relative to sgGal4 by a one-way ANOVA), and progressively decreased effects with increasing distances from enhancer center (Fig. 2f).

Together, these results not only establish the improved epigenetic editing systems for enhancer perturbation, but also demonstrate that the combinations of epigenetic modulators and transcriptional effector domains lead to more potent perturbations of locus-specific gene transcription. It is important to note that, although the constituent components of enCRISPRa (p300 and VP64) and enCRISPRi (LSD1 and KRAB) have been tested for transcriptional modulation of promoter and/or enhancer activity by fusing to dCas9 individually[16,20,40], the combinatorial effects on local epigenetic landscapes and gene transcription have not been examined previously. Therefore, the enCRISPRa and enCRISPRi systems that we describe here represent the first attempt to combine p300–VP64 and LSD1–KRAB in a single dCas9 complex for targeted modulation of enhancer function, respectively.

**Locus-specific epigenetic editing by enCRISPRi.** To determine the impact of enCRISPRi on epigenetic landscapes, we performed ChIP-seq analysis of dCas9, the enhancer-associated active histone marks (H3K4me1, H3K4me2 and H3K27ac), the repressive chromatin-associated H3K9me3 and H3K27me3, the hematopoietic lineage master TFs (GATA1 and TAL1), and CTCF. We also compared cells expressing nontargeting sgGal4 (control or C; Fig. 3), or sgHS2 with the single-effector dCas9-KRAB (K), dCas9-LSD1 (L) or the dual-effector enCRISPRi (LK or KL) (two replicate experiments for each ChIP-seq; Fig. 3, Supplementary Figs. 2–4; Supplementary Table 1).

Comparing to control (C), dCas9-LSD1 (L) and dCas9-KRAB (K), the dual-repressor enCRISPRi LK and KL resulted in more apparent loss of enhancer-associated activating histone marks H3K4me1 and H3K27ac at the targeted HS2 enhancer but not the other nontargeted genomic loci (Fig. 3; Supplementary Figs. 2 and 4). Both enCRISPRi LK and KL significantly increased the levels of the repressive H3K9me3 at the targeted HS2 enhancer but not the β-globin promoters. Of note, enCRISPRi also led to marked loss of H3K4me2 and H3K27ac at the β-globin gene-proximal promoters and gene bodies (Fig. 3; Supplementary Fig. 2), consistent with their transcriptional downregulation (Fig. 2b, c). These results suggest that enCRISPRi-mediated

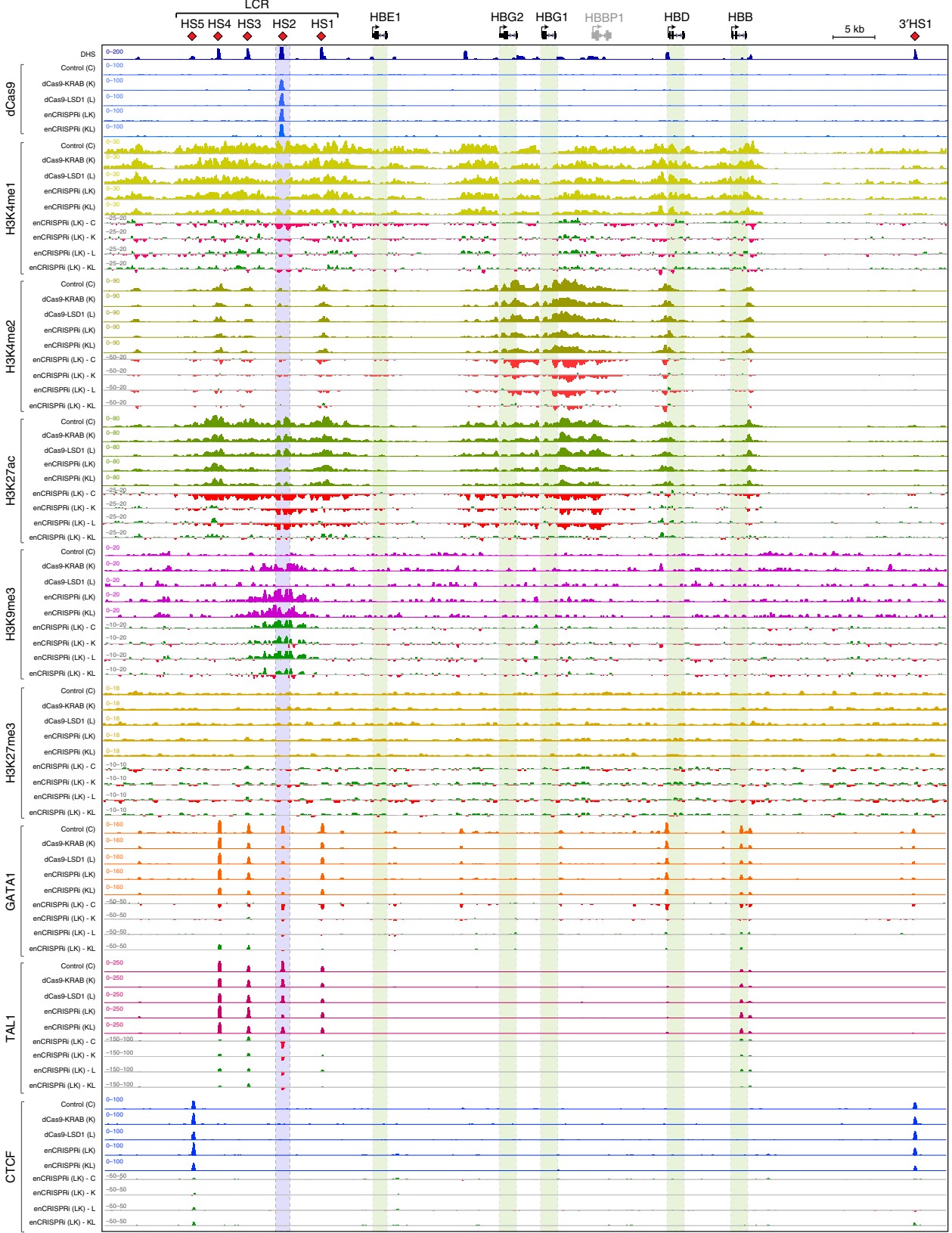

**Fig. 3 Locus-specific epigenetic reprogramming at the β-globin gene cluster.** Density maps are shown for ChIP-seq of dCas9, active histone marks (H3K4me1, H3K4me2, and H3K27ac), repressive H3K9me3 and H3K27me3, GATA1, TAL1, and CTCF at the β-globin cluster (chr11: 5,222,500−5,323,700; hg19) in K562 cells coexpressing nontargeting sgGal4 (control or C) or sgHS2 with dCas9-KRAB (K), dCas9-LSD1 (L) or enCRISPRi (LK and KL). Regions showing increased or decreased ChIP-seq signals in enCRISPRi (LK) relative to control, dCas9-KRAB, dCas9-LSD1 or enCRISPRi (KL) (enCRISPRi—C, enCRISPRi—K, enCRISPRi—L, or enCRISPRi—KL) are depicted in green and red, respectively. Blue bars denote the sgRNA-targeted HS2 enhancer. Green bars denote the β-globin genes. Independent replicate experiments are shown as rep1 and rep2 in Supplementary Figs. 2 and 3, respectively.

enhancer repression causes epigenetic changes at both targeted enhancers and associated gene promoters likely through enhancer–promoter interactions[41]. It is also important to note that, while dCas9-KRAB (K) led to significantly increased H3K9me3 at the targeted HS2 enhancer, dCas9-LSD1 (L) had no effect on H3K9me3 but instead decreased H3K4me1/2 (Fig. 3). These results are consistent with the roles of KRAB in promoting the formation of H3K9me3-mediated heterochromatin[42] and LSD1 in H3K4me1/2 removal[39]. Further, enCRISPRi (LK or KL) led to concurrent and more significant increases in H3K9me3 and decreases in H3K4me1/2 compared to dCas9-KRAB or dCas9-LSD1 alone (Fig. 3; Supplementary Figs. 2 and 5), indicating the cooperative or additive activity between two distinct repressor proteins. No significant enrichment of H3K27me3, the repressive histone marks catalyzed by Polycomb proteins[43], was detected at the β-globin gene cluster in K562 cells with or without dCas9-KRAB, dCas9-LSD1 or enCRISPRi (Fig. 3; Supplementary Fig. 2). Finally, while dCas9-KRAB or dCas9-LSD1 alone slightly or modestly affected the binding of GATA1 and TAL1, the key hematopoietic TFs required for the HS2 enhancer function[12,34,35], enCRISPRi led to further loss of GATA1 and TAL1 binding at the targeted HS2 enhancer (Fig. 3; Supplementary Figs. 3 and 5). Although the binding of dCas9 may interfere with TF–DNA interactions if the sgRNA-targeted sequences overlap with TF binding sites[44], the observed effects on GATA1/TAL1 binding are likely due to altered epigenetic landscapes at HS2 by dCas9 repressors. No significant effect on CTCF binding to the flanking insulators (HS5 and 3'HS1) was observed (Fig. 3; Supplementary Figs. 3 and 5).

Taken together, by focusing on the β-globin HS2 enhancer as a testbed, we provide evidence for the locus-specific epigenetic editing by enCRISPRi through KRAB-mediated H3K9me3 deposition and LSD1-mediated H3K4me1/2 removal. The dual effectors (KRAB and LSD1) act together to modulate locus-specific epigenetic modifications at the targeted enhancers and associated gene targets.

**Allele-specific perturbation of an oncogenic super-enhancer.** Having demonstrated the efficacy of the enCRISPRa and enCRISPRi perturbation systems in vitro, we examined whether we could modulate gene transcription and disease phenotypes in vivo by targeting disease-associated CREs. Recurrent mutations at an enhancer 8 kb upstream of the TAL1 proto-oncogene were discovered in T-cell acute lymphoblastic leukemia (T-ALL) cell lines and patients[45]. In each case, the heterozygous somatic mutations are acquired through insertion of variable number of nucleotides at the TAL1 enhancer sequences[45]. In Jurkat T-ALL cells, a heterozygous 12 bp insertion (GTTAGGAAACGG; Fig. 4a) introduces de novo binding motifs for the MYB proto-oncogene to initiate oncogenic super-enhancer (SE) formation[45]. To establish the proof-of-principle for enCRISPRa in dissecting the in vivo role of TAL1 oncogenic SE in T-ALL, we performed enCRISPRa-mediated activation of TAL1 enhancer in Jurkat cells. We designed two sgRNAs that specifically target the mutant (Mut) allele with protospacer-adjacent motifs (PAM) located within the 12 bp insertion sequence (sgMut1 and sgMut2; Fig. 4a). As controls, we designed two sgRNAs targeting both the wild-type (WT) and Mut enhancer sequences in the proximity of the 12 bp insertion sequence (sgWT1 and sgWT2). By stable coexpression of enCRISPRa and individual sgRNAs targeting the TAL1 oncogenic SE in Jurkat cells, we observed that sgMut1 and sgMut2 led to significant dCas9 binding to the Mut but not the WT allele, whereas sgWT1 and sgWT2 targeted dCas9 to both alleles (Fig. 4b). Further, in K562 cells lacking the Mut allele, no significant enrichment of K9 was observed at the WT alleles

upon coexpression of sgMut1 or sgMut2 (Fig. 4b), indicating the allele-specific targeting of enCRISPRa by sgMut1 and sgMut2. More importantly, we found that TAL1 mRNA and protein were significantly induced by sgRNAs targeting WT or Mut alleles relative to the sgGal4 control (Fig. 4c, d). Enhanced TAL1 expression by enCRISPRa led to significantly increased cell growth in vitro by PrestoBlue cell viability assay (Fig. 4e), consistent with the oncogenic role of TAL1 in T-ALL cell proliferation[45]. Since sgMut1 and sgMut2 specifically target the mutant enhancer sequences, these results suggest that allele-specific modulation of the mutant TAL1 SE allele achieves comparable efficacy on TAL1 transcriptional activation as targeting both alleles. Conversely, we found that enCRISPRi-mediated repression of TAL1 SE using the same sgRNAs resulted in significant downregulation of TAL1 mRNA and protein and impaired cell growth of Jurkat T-ALL cells (Fig. 4f–h).

Given the stronger effects on TAL1 activation or repression by sgMut2 and sgWT2 (Fig. 4c–h), we focused on these sgRNAs for the functional studies of enCRISPRa-mediated activation of TAL1 SE in T-ALL cells in vivo. Importantly, upon xenotransplantation into immunodeficient NSG (NOD-scid IL2Rg[null]) mice, enCRISPRa-mediated TAL1 enhancer activation led to greater tumor burden with significantly increased bioluminescence signals of the luciferase-expressing T-ALL cells in mice 4 weeks post transplantation (Fig. 4i, j). The mice transplanted with cells expressing TAL1 SE-targeting sgRNAs (sgMut2 and sgWT2) also displayed more severe leukemic phenotypes compared to the nontargeting sgGal4, resulting in increased infiltration of T-ALL leukemic cells in peripheral blood (PB) and bone marrow (BM) by flow cytometry and bloodsmear analyses (Fig. 4k, l). These results not only establish the functional role of the TAL1 oncogenic SE in promoting T-ALL development, but also demonstrate the efficacy of enCRISPRa and enCRISPRi for allele-specific perturbation of disease-associated enhancers in situ and in vivo.

**An inducible knock-in mouse model for in vivo enCRISPRi.** Systematic analysis of enhancer function in vivo remains a significant challenge. To explore the in vivo efficacy of enCRISPRi for enhancer perturbation, we engineered a new mouse model by site-specific knock-in (KI) of the dCas9-KRAB chimeric gene under the tetracycline-inducible promoter (TRE) into the Col1a1 locus, which was previously used as a "safe harbor" for robust transgene expression[46], in KH2 embryonic stem cells (ESCs) (Fig. 5a). KH2 ESCs also harbor the Rosa26-M2rtTA KI allele allowing expression of the rtTA-M2 transactivator for doxycycline-inducible studies[47]. After blastocyst injection of targeted ESCs and screening of germline transmitted offsprings, the founder mice were crossed to obtain the dCas9-KRAB::Rosa26-M2rtTA heterozygous or homozygous KI mice (named dCas9-KRAB KI hereafter) (Fig. 5b). The inducible dCas9-KRAB protein expression was confirmed in the targeted KH2 ESCs and bone marrow cells isolated from the dCas9-KRAB KI mice (Fig. 5c). To determine the effect of dCas9-KRAB expression on mouse hematopoietic development, we performed complete blood counts of peripheral blood and flow cytometry of various mature hematopoietic cell types including erythroid (Ter119+), B-lymphoid (B220+), T-lymphoid (CD3+) and myeloid (Mac1 + Gr1+) cells in bone marrow and spleen of wild-type (WT) or dCas9-KRAB KI mice with or without Dox treatment, respectively. Our results revealed no overt abnormalities on the frequency of mature hematopoietic cells before or after Dox-induced dCas9-KRAB expression (Supplementary Fig. 6a–c). Furthermore, the cellularity and frequency of various hematopoietic stem/progenitor cell populations in bone marrow and spleen were

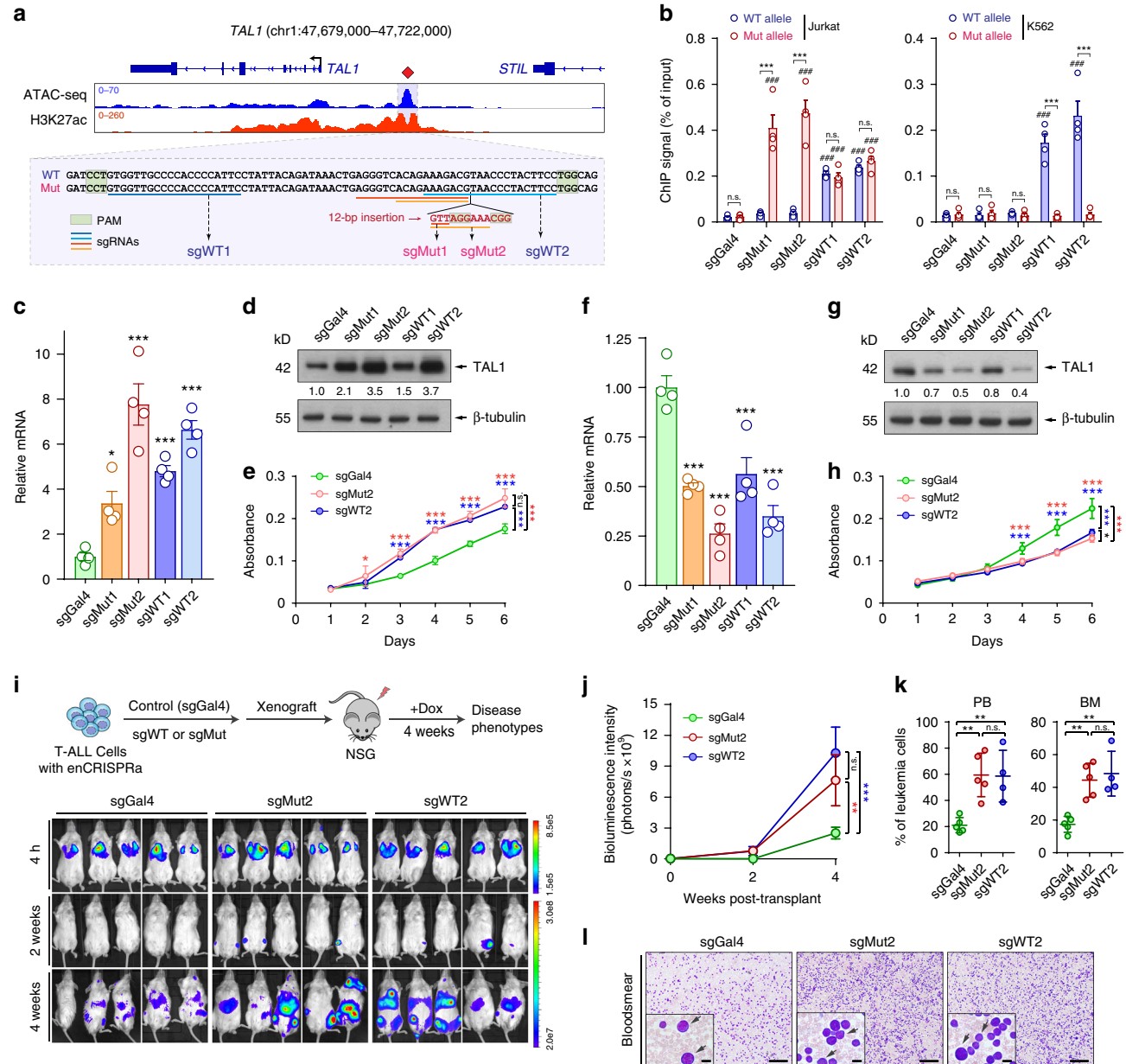

**Fig. 4 Perturbations of *TAL1* oncogenic super-enhancer in xenografts. a** Density maps are shown for ATAC-seq and H3K27ac ChIP-seq at *TAL1* (chr1:47,679,000–47,722,000; hg19) in Jurkat cells. The *TAL1* oncogenic SE is shown as blue shaded lines. The positions of sgRNAs and PAM sequences are shown as colored lines and boxes, respectively. **b** Chromatin occupancy of dCas9-p300 by target-specific (sgMut1, sgMut2, sgWT1 and sgWT2) or nontargeting sgGal4 in Jurkat or K562 cells. ChIP signals (% of input) are shown as mean ± SEM ($n = 4$ experiments). **c** Expression of *TAL1* mRNA in Jurkat cells upon enCRISPRa-mediated enhancer activation. Results are mean ± SEM ($n = 4$ experiments). **d** Expression of TAL1 protein upon enCRISPRa-mediated enhancer activation. The quantified TAL1 expression is shown. **e** Activation of *TAL1* SE promoted T-ALL growth in vitro. Relative absorbance by cell viability assays (y axis) at different days of culture (x axis) is shown ($n = 3$ experiments). **f** Expression of *TAL1* upon enCRISPRi-mediated enhancer repression in Jurkat cells. Results are mean ± SEM ($n = 4$ experiments). **g** Expression of TAL1 protein upon enCRISPRi-mediated enhancer repression. **h** Repression of *TAL1* SE impaired T-ALL growth in vitro ($n = 3$ experiments). **i** Activation of *TAL1* SE promoted T-ALL growth in NSG mice xenografted with Jurkat cells transduced with sgGal4, sgMut2 or sgWT2, respectively. Bioluminescence intensity is shown at 4 h, 2 and 4 weeks post transplantation. **j** Quantification of bioluminescence intensity is shown. Results are mean ± SEM ($n = 5$ recipients per group). **k** Frequencies of leukemia cells in BM and PB of xenografted NSG mice 4 weeks post transplantation. Results are mean ± SEM ($n = 5$, 5, and 4 recipients for sgGal4, sgMut2 and sgWT2, respectively). **l** Representative bloodsmear images of NSG mice 4 weeks post transplantation. The inset images indicate the zoom-in view. Representative leukemia cells are indicated by arrowheads. Scale bars, 200 and 20 μm for full and insert images, respectively. Results are mean ± SEM and analyzed by a one-way or two-way ANOVA. *$P < 0.05$, **$P < 0.01$, ***$P < 0.001$, ###$P < 0.001$, n.s. not significant. Source data are provided as a Source Data file.

comparable in WT and KI mice with or without Dox treatment (Supplementary Fig. 6d–f), indicating that the inducible dCas9-KRAB expression does not affect normal blood development.

**In vivo functional interrogation of lineage-specific enhancers.** Hematopoiesis serves as a paradigm for understanding stem cell differentiation controlled by lineage-specifying TFs[48]. The self-renewing hematopoietic stem cells (HSCs) give rise to all mature

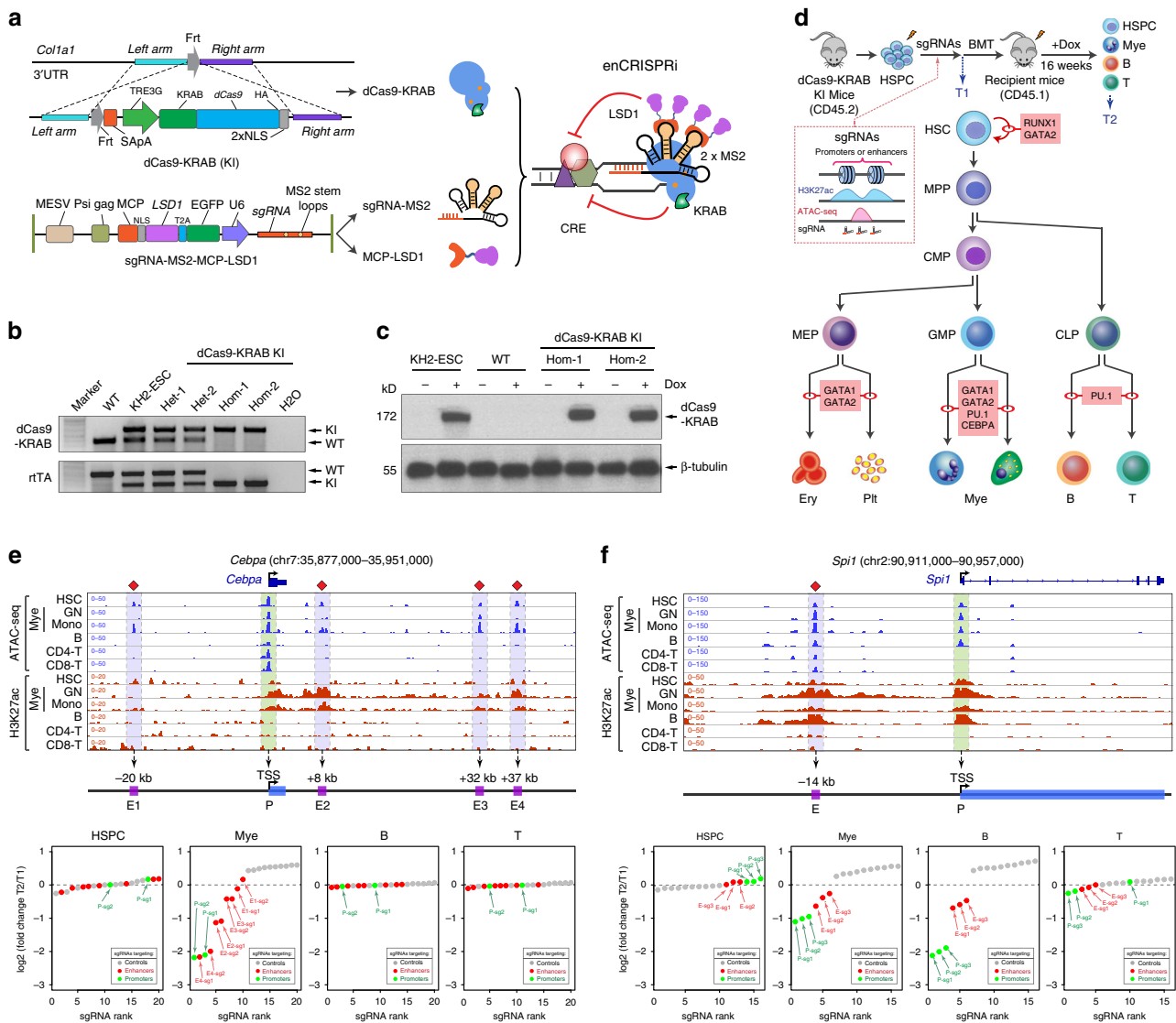

**Fig. 5 Locus-specific in vivo enhancer perturbation. a** Schematic of site-specific KI of tetracycline-inducible dCas9-KRAB into the *Col1a1* locus. Coexpression of dCas9-KRAB, sgRNA-MS2 and MCP-LSD1 leads to assembly of enCRISPRi complex in vivo. **b** Validation of dCas9-KRAB and rtTA KI or WT alleles by genotyping PCR. C57BL/6 WT mouse and targeted KH2-ESC were used as controls. Two independent heterozygous (Het) and homozygous (Hom) KI mice were analyzed. **c** Dox-inducible expression of dCas9-KRAB fusion protein was confirmed by Western blot in the targeted KH2-ESC and two independent dCas9-KRAB KI mice. β-tubulin was analyzed as the loading control. **d** Schematic of in vivo perturbation of lineage-specific enhancers in dCas9-KRAB KI mice. **e** In vivo enCRISPRi perturbation of *Cebpa* CREs revealed lineage-specific requirement of *Cebpa* enhancers during hematopoiesis. Waterfall plots are shown for target-specific sgRNAs (green and red dots) and nontargeting control sgRNAs (gray dots) by the mean normalized log2 fold changes in HSPCs, myeloid, T or B cells 16 weeks post BMT (T2) relative to pooled sgRNA-transduced HSPCs (T1) from two independent replicate screens (*n* = 3 recipient mice per screen). Density maps are shown for ATAC-seq and H3K27ac ChIP-seq at the *Cebpa* locus (chr7:35,877,000–35,951,000; mm9) in bone marrow HSC, granulocytes (GN), monocytes (Mono), B, CD4+ and CD8+ T cells, respectively. The annotated *Cebpa* promoter (P) and enhancers (E1 to E4) are indicated by green and blue shaded lines. Results from independent replicate screens and statistical analyses are shown in Supplementary Fig. 7a, b. **f** In vivo enCRISPRi perturbation of *Spi1* CREs during hematopoiesis. Density maps are shown for ATAC-seq and H3K27ac ChIP-seq at the *Spi1* locus (chr2:90,911,000–90,957,000; mm9) in bone marrow HSC, GN, Mono, B, CD4+ and CD8+ T cells, respectively. The annotated *Spi1* promoter (P) and enhancer (E) are indicated by green and blue shaded lines. Results from independent replicate screens and statistical analyses are shown in Supplementary Fig. 7c, d. Source data are provided as a Source Data file.

blood cell lineages through a hierarchy of progenitors during lineage specification. In the bone marrow transplantation (BMT) setting, HSCs are capable of reconstituting the entire blood system of a recipient, whereas the short-lived progenitors do not. HSC self-renewal and/or lineage determination are controlled by a small number of TFs, many of which function in a highly lineage-specific manner and are regulated by tissue-specific and/or developmentally regulated enhancers[48]. We reasoned that by epigenetic modulation of lineage-specific enhancers in HSCs

followed by BMT, we could assay the HSC-derived mature cell lineages as the "readout" for the functional impact of enhancer perturbations during HSC lineage differentiation in vivo.

To this end, we devised an in vivo enhancer perturbation assay by combining the dCas9-KRAB KI mice with sgRNA-MS2 and MCP-LSD1 to assemble the enCRISPRi complex in vivo (Fig. 5a). We then determined the functional role of lineage-specific enhancers associated with major hematopoietic TFs by in vivo enCRISPRi (Fig. 5d). We focused on five TFs including *Cebpa*[49],

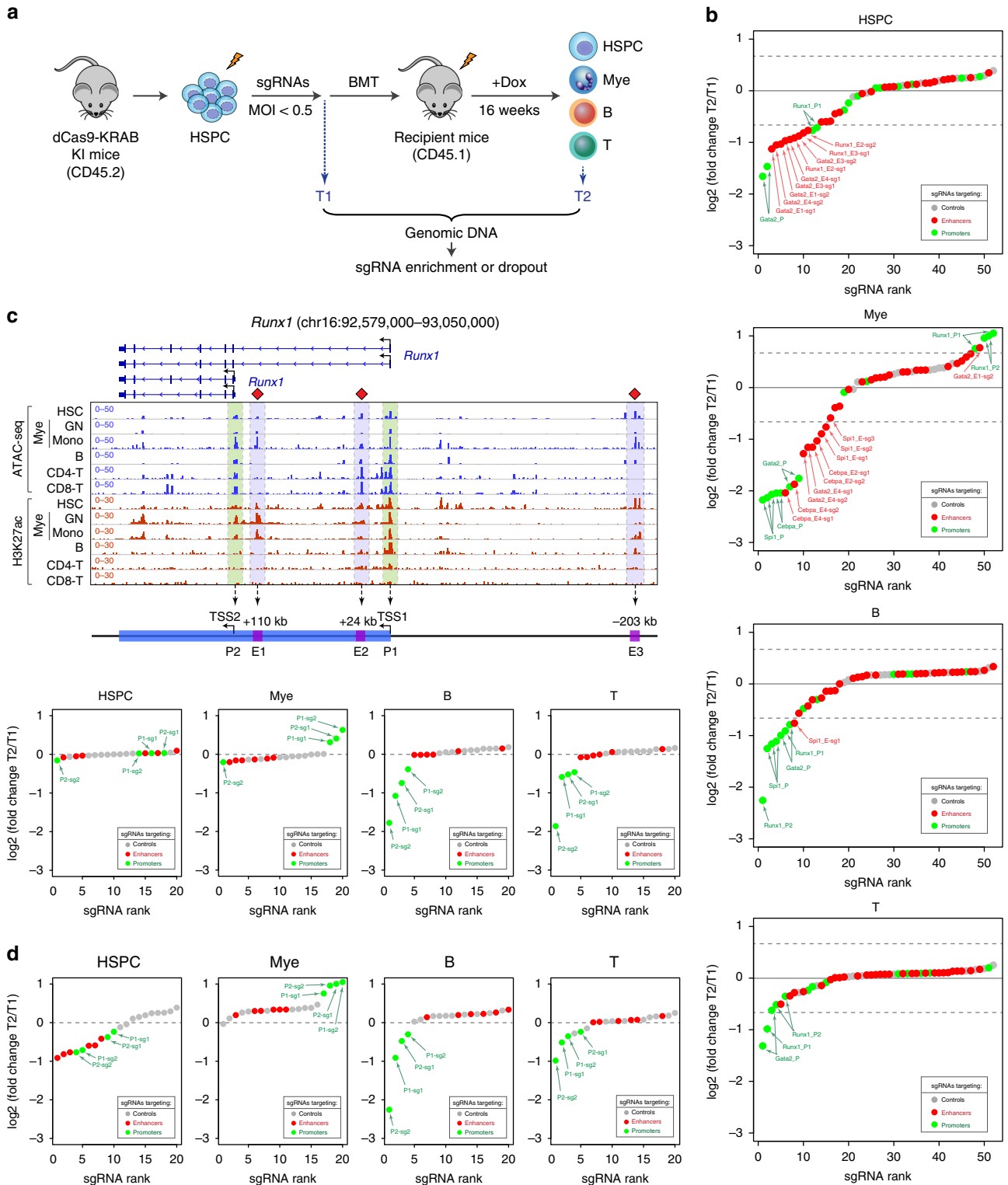

*Spi1* (or PU.1)[50], *Gata1*[51], *Gata2*[52] and *Runx1*[53] that play critical roles in HSC function and/or lineage differentiation[48]. Each gene contains one or multiple annotated enhancers based on chromatin accessibility by ATAC-seq and enhancer-associated H3K27ac by ChIP-seq[54] (Figs. 5e, f, 6c; Supplementary Figs. 7–9). We designed 2 or 3 sgRNAs for each enhancer, 2 or 3 sgRNAs for each gene promoter as positive controls, and 10 nontargeting sgRNAs as negative controls (Supplementary Table 2). The target-specific and control sgRNAs were pooled for each gene for

locus-specific enhancer perturbation (total 5 pools with 16–20 sgRNAs in each pool). CD45.2+ BM lineage negative hematopoietic stem/progenitor cells (HSPCs) from dCas9-KRAB KI mice were isolated and retrovirally transduced with pooled sgRNAs at MOI ≤ 0.5 to ensure that each cell contained no more than one sgRNA. The transduced cells were transplanted into CD45.1+ lethally irradiated recipient mice, followed by Dox administration to induce dCas9-KRAB expression in recipient mice for 16 weeks. By this time, all donor-derived hematopoietic

**Fig. 6 Multiloci perturbations of developmental enhancers during hematopoiesis. a** Schematic of in vivo pooled sgRNA-based multiloci perturbations of developmentally regulated enhancers in dCas9-KRAB KI mice. **b** In vivo perturbation of annotated CREs for five key hematopoietic TFs revealed the functional requirement of lineage-specific enhancers for HSC differentiation to one or multiple hematopoietic lineages. Waterfall plots are shown for target-specific sgRNAs (green and red dots) and nontargeting control sgRNAs (gray dots) by the mean normalized log2 fold changes in HSPCs, myeloid, T or B cells 16 weeks post BMT (T2) relative to pooled sgRNA-transduced HSPCs (T1) from two independent replicate screens ($n = 15$ recipient mice per replicate screen). Results from independent replicate screens and statistical analyses are shown in Supplementary Fig. 12. **c** In vivo enCRISPRi of *Runx1* CREs during hematopoiesis by single locus-based perturbation. Density maps are shown for ATAC-seq and H3K27ac ChIP-seq at the *Runx1* locus (chr16:92,579,000–93,050,000; mm9) in bone marrow HSC, GN, Mono, B, CD4+ and CD8+ T cells, respectively. The annotated *Runx1* promoters (P1 and P2) and enhancers (E1–E3) are indicated by green and blue shaded lines. Results from independent replicate screens and statistical analyses are shown in Supplementary Fig. 9a, b. **d** In vivo enCRISPRi of *Runx1* CREs during hematopoiesis by multiplexed perturbation. Results from independent replicate screens and statistical analyses are shown in Supplementary Fig. 9c, d.

cells (CD45.2+) were differentiated from repopulating donor HSCs instead of short-lived progenitors. We collected cells before BMT (T1) as the baseline control and donor-derived BM HSPCs and mature myeloid (Gr1+Mac1+), B (B220+) and T (CD3+) lymphoid cells in the peripheral blood of recipient mice 16 weeks post BMT (T2). We did not analyze erythroid and megakaryocytic lineages because mature erythroblasts and platelets do not contain nuclei. Genomic DNA from T1 and T2 cells were isolated to amplify the sgRNA sequences, followed by high-throughput sequencing. If enCRISPRi-mediated repression of an enhancer or promoter impaired its function and target gene expression, the corresponding sgRNAs would be depleted (or enriched) in T2 relative to T1 cells (Fig. 5d).

We performed five independent locus-specific enhancer perturbation screens, each with two independent replicate experiments, for *Cebpa*, *Spi1*, *Gata1*, *Gata2* and *Runx1*, respectively (Figs. 5e, f, 6c; Supplementary Figs. 7–9). Deletion of *Cebpa* or its downstream enhancer (E4, +37 kb from TSS; Fig. 5e) led to defective myeloid-lineage priming and differentiation without affecting lymphopoiesis[49,55]. Consistent with these findings, sgRNAs for E4 enhancer were the top depleted sgRNAs in myeloid cells (ranked #2 and #4) next to the positive control sgRNAs for the *Cebpa* promoter (ranked #1 and #3; Fig. 5e) in independent screen experiments (Supplementary Fig. 7a, b). More importantly, our results revealed that *Cebpa* E2 (+8 kb) enhancers was also required for in vivo myelopoiesis, whereas E1 (−20 kb) and E3 (+32 kb) were dispensable. None of the *Cebpa* enhancers was required for HSPCs, B- and T-cell development by considering both the fold changes of sgRNA abundance and the significance of sgRNA depletion between T1 and T2 (Fig. 5e; Supplementary Fig. 7a, b).

For *Spi1* locus, we observed significant depletion of all three sgRNAs for its −14kb enhancer in myeloid and B but not T cells (Fig. 5f; Supplementary Fig. 7c, d). These results are consistent with the role of Spi1 for normal myeloid and B-cell development[50,56], and validate the in vivo requirement of its upstream enhancer in regulating myelopoiesis and B lymphopoiesis[57]. With regards to *Gata1*, while sgRNAs for *Gata1* promoters were modestly depleted in myeloid cells consistent with the role of GATA1 in eosinophil and mast cell function[48,58], none of the sgRNAs for *Gata1* enhancers were depleted in any of the three lineages (Supplementary Fig. 8a, b). Interestingly, *Gata1* enhancers displayed chromatin accessibility and H3K27ac enrichment in HSCs but not mature myeloid, B and T cells, suggesting that the HSC-specific *Gata1* enhancers are not required for HSC differentiation to mature myeloid or lymphoid lineages. For *Gata2*, we observed that an intronic enhancer (+9.5 kb, E4) was required for in vivo myelopoiesis, whereas the other three gene-distal enhancers were minimally required for any of the three lineages (Supplementary Fig. 8c, d).

The lack of sgRNA depletion by targeting enhancers or promoters could be due to inefficient chromatin targeting of

enCRISPRi or nonessential roles of the targeted CREs in gene expression. To distinguish between these possibilities, we measured target gene expression and dCas9 chromatin occupancy in BM HSPCs by qRT-PCR and ChIP (Supplementary Figs. 10 and 11), respectively. It is important to note that the in vivo functional impacts of enhancer perturbation largely correlated with the levels of target gene repression by enCRISPRi at individual enhancers or promoters in HSPCs (Supplementary Fig. 10). The lineage-restricted impacts on HSC differentiation also correlated with the presence of enhancer-associated chromatin features (ATAC-seq and H3K27ac) in different lineages (Figs. 5e, f, 6c; Supplementary Figs. 7–9), highlighting the context-specific requirements of developmental enhancers in regulating gene expression and lineage differentiation. Further, all the tested target-specific sgRNAs resulted in significant and comparable dCas9 binding at the targeted enhancers or promoters relative to nontargeting sgGal4 (Supplementary Fig. 11), suggesting that the lack of target gene repression is unlikely due to inefficient dCas9/sgRNA targeting. It is important to note that the presence of dCas9 chromatin occupancy may not reflect the repressive function of the assembled enCRISPRi complexes on chromatin; thus, our findings do not completely exclude the possibility of inefficient enCRISPRi-mediated repression at some of the targeted CREs.

**Multiloci perturbations of enhancers during hematopoiesis.** Locus-specific perturbation screens provided information for the "essentiality" of each enhancer within its local context, but did not reveal the relative importance of enhancers across multiple loci during hematopoiesis. To address this question, we pooled the target-specific and control sgRNAs for all five hematopoietic TFs, and performed multiloci enhancer perturbation screens (Fig. 6a). We observed that the top depleted enhancer sgRNAs in myeloid cells included *Cebpa* +37 kb (E4) and +8 kb (E2) enhancers, *Gata2* +9.5 kb (E4) and *Spi1* –14 kb enhancers when considering the fold changes of sgRNA abundance and the significance of sgRNA depletion (Fig. 6b; Supplementary Fig. 12). The sgRNAs for *Spi1* and *Gata2* enhancers were modestly or slightly depleted in B and T cells, respectively. Moreover, the sgRNAs for *Gata2* E1, E3 and E4 enhancers and *Runx1* E2 and E3 enhancers were modestly or slightly depleted in HSPCs (Fig. 6b and Supplementary Fig. 12). These results illustrate the disparate requirements for distinct developmental enhancers during hematopoiesis, with some enhancers broadly required for multiple lineages (e.g. *Gata2* or *Spi1* enhancers) while others uniquely required for one specific lineage (e.g. *Cebpa* enhancers). Our results also demonstrate that the in vivo enhancer functions cannot be solely predicted based on enhancer-associated epigenetic features. For example, while the annotated *Cebpa* enhancers share similar chromatin accessibility and H3K27ac enrichment in HSCs and/or myeloid cells, the enCRISPRi-mediated repression

of the +8 kb E2 and +37 kb E4 enhancers resulted in more profound impacts on myeloid development (Figs. 5e and 6b).

By locus-specific enhancer perturbation, we found that sgRNAs for *Runx1* promoters (P1 and P2) were significantly depleted in B and T cells but enriched in myeloid cells, whereas none of sgRNAs for the annotated *Runx1* enhancers showed differential enrichment (Fig. 6c and Supplementary Fig. 9a, b). Similar results were replicated in multiloci perturbations (Fig. 6d and Supplementary Fig. 9c, d). These findings demonstrate that enCRISPRi-mediated repression of *Runx1* promoters impaired HSC differentiation to B and T lineages but promoted myeloid differentiation; however, enCRISPRi-mediated enhancer repression was ineffectual despite efficient enCRISPRi targeting (Supplementary Fig. 11d), supporting the nonessential roles of *Runx1* enhancers in myelopoiesis or lymphopoiesis. More importantly, the opposing phenotypes of repressing *Runx1* by enCRISPRi in myeloid vs. lymphoid lineages are consistent with the differential roles of *Runx1* in hematopoiesis[59]. Hematopoietic cell-specific *Runx1* KO in mice led to the development of myeloproliferative phenotypes characterized by defective B- and T-cell maturation, and increased myelopoiesis[59]. Our results faithfully recapitulated the phenotypic manifestations of *Runx1* deficiency, illustrating the utility of enCRISPRi-based epigenetic editing in functional analysis of *cis*-regulatory elements during in vivo development. Hence, the exemplified applications of enhancer perturbations in vitro (Figs. 1–3), in xenografts (Fig. 4) and in vivo (Figs. 5 and 6) in multiple cell models establish the enhancer-targeting CRISPR epigenetic editing systems as general tools for functional interrogation of noncoding regulatory genome in development and disease.

## Discussion

Here we describe the dual-effector-containing CRISPR epigenetic editing systems for locus-specific activation or repression of transcriptional enhancers or other CREs in native chromatin. Using epigenetic writer proteins (p300 and LSD1) that specifically modulate histone modifications associated with active enhancers, the enCRISPRa and enCRISPRi systems enable efficient interrogation of enhancer function in human and mouse cells in vitro, in xenografts and in vivo. Although the concept of engineering dual or multi-effector-based dCas9 complexes by dCas9 fusions or MS2-MCP scaffolding has been previously described[19,24,27,29,38], the strategies combining enhancer-targeting epigenetic modulators and transcriptional effector domains (e.g. VP64 and KRAB) are less explored. Furthermore, the in vivo efficacy of dCas9-based epigenetic editing in developmental gene regulation and/or disease phenotypes remains largely unknown[30,31]. In this study, we demonstrate that the combinations of independent repressors (LSD1 and KRAB in enCRISPRi) or activators (p300 and VP64 in enCRISPRa) lead to more potent modulations of gene transcription by remodeling epigenetic landscapes at the targeted genomic loci. KRAB is associated with heterochromatin formation[42], whereas LSD1 removes enhancer-associated H3K4me1/2[39]. The combined effects on H3K9me3 deposition and H3K4me1/2 removal exceeded individual effectors when targeted to the HS2 enhancer (Fig. 3; Supplementary Figs. 2, 3 and 5), suggesting that different effectors act together for maximal enhancer perturbations. Moreover, the enCRISPRi and enCRISPRa systems permit parallel analyses of the same loci by both loss- and gain-of-function approaches, thus facilitating the identification of CREs that are necessary and/or sufficient for target gene expression. It is important to note that, while our studies focus on the applications of enCRISPRi and enCRISPRa for modulating enhancer activities, the dual-effector systems also work effectively on modulating

gene expression when targeted to gene-proximal promoters in vitro and in vivo (Figs. 1b, 5e, f and 6b−d; Supplementary Figs. 7–9 and 12).

We found that a single sgRNA targeting a single enhancer was sufficient for gene repression; however, the position of sgRNAs can significantly affect the effectiveness of CRISPR epigenetic editing as shown in the case of HS2 enhancer (Fig. 2e, f). Given that we positioned sgRNAs based on the distances to the DHS or ATAC-seq peak summits at enhancers, these results indicate that efficient enhancer perturbations require maximal interference with the chromatin binding of enhancer-regulating TFs. Consistent with this idea, the dual-effector enCRISPRi led to more significant disruption of GATA1 and TAL1 binding than single-effector dCas9-KRAB or dCas9-LSD1 at HS2 enhancer (Fig. 3). These findings are distinct from CRISPRi in promoter repression, in which targeting sgRNAs to promoter downstream sequences (50–100 bp) was more effective in gene repression likely due to the interference with transcription initiation and/or elongation complexes[17]. In future studies, high-resolution tiling screens of critical TF binding site(s) at enhancer sequences may elicit the functional roles of individual TF binding sites in enhancer regulation. Such studies will not only provide insights into the regulatory mechanisms for enhancer function, but also identify selective "vulnerabilities" of enhancers such as functionally relevant enhancers within the super-enhancer cluster and/or specific TF binding sites within enhancers that may be employed to precisely control gene expression.

Furthermore, there are important considerations for further improvements of the CRISPR epigenetic editing systems including potential therapeutic applications. First, the potential off-target effects due to nonspecific dCas9 binding need to be evaluated using independent sgRNAs, ChIP-PCR or ChIP-seq analysis of dCas9 chromatin occupancy, expression analysis of the targeted and nontargeted genes, and other orthogonal approaches. Second, the effect of steric hindrance of dCas9 binding should be considered when designing CRISPR epigenetic editing assays. Third, the expression of the targeted genes upon activation by dCas9 activator complexes needs to be determined to minimize nonphysiological overexpression for certain studies including potential therapeutic applications. Fourth, given the large sizes of the dCas9-effector fusion proteins, improved methods are needed for efficient delivery of the dCas9-based epigenetic editing reagents to targeted cell types. Finally, the long-term and genome-wide effects of prolonged epigenetic editing in targeted cells should be evaluated.

By in vivo perturbation screens, we identified several candidate enhancers required for lineage differentiation of HSCs (Figs. 5 and 6; Supplementary Figs. 7–9 and 12). Although the lineage-specific essentiality largely correlated with the presence of enhancer-associated chromatin accessibility and H3K27ac, the functional roles of individual enhancers cannot be reliably predicted based on only epigenetic features. For instances, while the annotated *Cebpa* enhancers share similar chromatin features, only +8 kb E2 and +37 kb E4 enhancers are indispensable for *Cebpa* expression and myeloid development (Figs. 5e, 6b and Supplementary Fig. 7). Similarly, while several annotated *Runx1* enhancers harbor enhancer-associated chromatin features in HSCs and/or differentiated lineages, none of them was required for *Runx1* expression or function in HSC differentiation to myeloid and lymphoid cells in vivo (Fig. 6; Supplementary Figs. 9 and 12). These results highlight the importance of analyzing enhancer function in situ by loss- and gain-of-function assays, such as the enCRISPRi and enCRISPRa epigenetic editing systems described here, during in vivo development. These assays require the analysis of cellular phenotypes such as stem cell differentiation, cell proliferation, viability, and/or response to stimuli as readouts to quantify the functional impacts.

Therefore, the development of dCas9-KRAB KI mouse model provides opportunities for in vivo functional interrogation of enhancers and other CREs in lineage differentiation of tissue stem cells. Simultaneous analysis of many genes or loci allows efficient functional screens to prioritize relevant CREs for in-depth characterization. In future studies, the inducible dCas9-KRAB KI mouse model combined with various disease models will provide new insights into the roles of noncoding CREs in disease pathogenesis in vivo. In addition, the improved CRISPR epigenetic editing systems should accelerate functional follow-up studies of disease or trait-associated genetic variants and cancer-associated somatic alterations, many of which reside in noncoding CREs including enhancers. Finally, the enhanced CRISPR epigenetic editing may suggest potential therapeutic strategies by generating targeted epigenetic modifications to alter the expression of desired genes. Hence, the dual-activator or repressor-containing CRISPR perturbation systems provide additional tools for the CRISPR toolbox for functional analysis of noncoding regulatory elements in situ and in vivo. Our studies not only identify candidate lineage-specific enhancers required for hematopoiesis, but also establish widely applicable platforms for unbiased analysis of noncoding regulatory genome which can be extended to other cell types and human diseases.

## Methods

**Cells and cell culture.** Human K562 (ATCC #CCL-243) and Jurkat (ATCC #TIB-152) cells were obtained from ATCC and cultured in RPMI 1640 medium containing 10% fetal bovine serum (FBS), 1% penicillin/streptomycin (P/S). Human HEK293T (ATCC #CRL-11268) cells were obtained from ATCC and cultured in DMEM medium containing 10% FBS and 1% P/S. KH2 ESCs were obtained from Dr. Stuart Orkin's laboratory at Boston Children's Hospital and cultured in DMEM containing 10% ES-certified FBS (GemCell™, Cat# 100-500), 1% P/S, 2 mM L-Glutamine (Gibco), 0.1 mM MEM nonessential amino acids solution (Gibco), 1 mM sodium pyruvate (Gibco), 0.1 mM β-Mercaptoethanol and 1000 U/ml recombinant mouse LIF (ESGRO, Cat# ESG1107). All cultures were incubated at 37 °C in 5% $CO_2$. No cell line used in this study was found in the database of commonly misidentified cell lines that is maintained by ICLAC and NCBI Bio-Sample. All cell lines were tested for mycoplasma contamination.

**Plasmids.** To generate the inducible dCas9-p300 expression vector, the p300 HAT core domain (p300 core) was PCR amplified from the pcDNA-dCas9-p300-Core vector (Addgene, Plasmid #61357) and cloned into MluI/BstXI digested pHR-TRE3G-KRAB-dCas9-P2A-mCherry backbone, which was a gift from Luke A. Gilbert[23]. To generate the inducible dCas9-LSD1 expression vector, LSD1 open-reading frame (ORF) was amplified and cloned into the pHR-TRE3G-KRAB-dCas9-P2A-mCherry to replace the KRAB domain. To generate the enCRISPRa sgRNA vector, the MCP-VP64-IRES-mCherry cassette was PCR amplified from the pJZC34 vector (Addgene, plasmid #62331) and cloned into BsrGI/EcoRI digested lenti-sgRNA (MS2)-zeo backbone (Addgene, plasmid # 61427). Then the mCherry cassette was replaced with zsGreen1 by In-Fusion® HD Cloning Kit (Clontech). To generate the enCRISPRi-LK sgRNA vector, the KRAB sequence was PCR amplified from the pLV hU6-sgRNA hUbC-dCas9-KRAB-T2A-Puro vector (Addgene, plasmid #71236) and cloned into the enCRISPRa sgRNA vector to replace VP64. To generate the enCRISPRi-KL sgRNA vector, MCP-LSD1 was amplified and cloned into pMLS-NRAS-T2A-GFP-polyA-U6 to replace NRAS. To generate the MCP fusions, p300, LSD1 or VP64 were amplified and cloned into Lenti-MS2-P65-HSF1-Hygro (Addgene, plasmid #61426) to replace P65-HSF1. To generate Lenti-sgRNA-MS2-zsGreen1, zsGreen1 was amplified and cloned into Lenti-sgRNA (MS2)-zeo (Addgene #61427) to replace zeocin. Lenti-dCas9-VP64-Blast (Addgene #61425),3xFLAG-dCas9/pCMV-7.1 (Addgene #47948) and LentiGuide-Puro (Addgene #52963) were obtained from Addgene.

**Design and cloning of sgRNAs.** sgRNAs were designed to minimize off-targets based on publicly available filtering tools (http://crispr.genome-engineering.org/crispr/). Briefly, oligonucleotides were annealed in the following reaction: 10 μM guide sequence oligo, 10 μM reverse complement oligo, T4 ligation buffer (1×), and 5U of T4 polynucleotide kinase (NEB) with the cycling parameters of 37 °C for 30 min; 95 °C for 5 min and ramp down to 25 °C at 5 °C/min. The annealed oligos were cloned into the sgRNA vectors using a Golden Gate Assembly strategy including: 100 ng of circular sgRNA vector plasmid, 0.2 μM annealed oligos, buffer 2 (1×) (NEB), 20 U of BbsI or BsmBI restriction enzyme, 0.2 mM ATP, 0.1 mg/ml BSA, and 750 U of T4 DNA ligase (NEB) with the cycling parameters of 20 cycles at 37 °C for 5 min, 20 °C for 5 min, followed by 80 °C incubation for 20 min. Insertion of sgRNA was validated by Sanger sequencing. Lentiviruses containing

sgRNAs were packaged in HEK293T cells. Briefly, 2 μg of psPAX2, 1 μg of pMD2.G and 5 μg sgRNA vectors were cotransfected into HEK293T cells seeded in 10 cm petri dish. Lentiviruses were harvested from the supernatant 48−72 h post transfection. Dox-inducible enCRISPRi or enCRISPRa cell lines were then transduced with sgRNA-expressing lentiviruses in six-well plates. To maximize sgRNA expression, top 1–5% of GFP-positive cells were FACS sorted 48 h post infection. The sequences for all sgRNAs are listed in Supplementary Table 2.

**Tetracycline-inducible dCas9-KRAB knock-in mouse model.** Site-specific knock-in (KI) of tetracycline-inducible dCas9-KRAB transgene was generated by flippase (FLPe)-mediated recombination as previously described[34,60]. KH2 mouse embryonic stem cells (ESCs) harboring the M2rtTA tetracycline-responsive transactivator in Rosa26 locus and an engineered Col1a1 locus with an frt site and ATG-less hygromycin resistance gene were used[60]. A targeting construct pBS31-TRE-dCas9-KRAB containing the PGK promoter, an frt site, a tetracycline-inducible minimal CMV promoter, the dCas9-KRAB transgene, and an ATG initiation codon was coelectroporated with the pCAGGS-FLPe-puro into KH2 ESCs at 500 V and 25 μF using a Gene Pulser II (Bio-Rad). Cells were selected with hygromycin (140 μg/ml), and positive clones were expanded and analyzed by genotyping PCR. Targeted ESC clones were injected to embryonic day 3.5 mouse blastocysts to obtain the founder mice. Chimeric founder mice were bred with C57BL/6 mice, and offsprings with germline transmission were genotyped using primers in Supplementary Table 2 and intercrossed to generate dCas9-KRAB heterozygous or homozygous KI mice. All mouse experiments were performed under protocols approved by the Institutional Animal Care and Use Committee of UT Southwestern Medical Center (UTSW).

**Xenograft experiments.** Luciferase cassette was amplified and cloned into pLVX-Puro vector (Clontech, Catalog No. 632164). Lentivirus was produced to transduce Jurkat cells coexpressing enCRISPRa with control sgGal4, sgWT2 or sgMut2, respectively. Puromycin selection (1 μg/ml) was performed 3 days after infection. Six- to eight-week-old female NOD-SCID (NSG) mice were sublethally irradiated (2.5 Gy) half day before the transplantation. Cells ($1 \times 10^6$/mice) were resuspended in PBS (200 μl/mice) and intravenously transplanted. Transplanted mice underwent in vivo bioluminescence imaging at various time points to evaluate tumor growth. Briefly, following intraperitoneal injection of 150 mg/kg D-luciferin (Gold Biotechnology), mice were imaged, and bioluminescence intensity was quantitated using Living Image 3.2 acquisition and analysis software (Caliper Life Sciences). Total flux values were determined by drawing regions of interest (ROI) of identical size over each mouse and were presented in photons (p)/second (s). The initial measurement at 4 h post xenograft was performed to confirm the successful xenotransplantation of the luciferase-expressing leukemia cells. As the surviving leukemia cells proliferate over time, the bioluminescence signals increase in subsequent measurements (weeks 2 and 4). Four weeks after transplantation, the peripheral blood, bone marrow and spleen were assessed for engraftment by flow cytometry. Bloodsmear was performed and stained with May−Grunwald−Giemsa as previously described[61]. The blue stained cells indicated the circulating leukemia cells.

**Generation of enCRISPRi or enCRISPRa cell lines.** To generate inducible enCRISPRi and enCRISPRa stable cell lines, the target cells were transduced with lentivirus expressing dCas9-effector fusion proteins with mCherry (dCas9-KRAB, dCas9-LSD1 or dCas9-p300) and the Tet-On 3G rtTA transactivator with BFP (Supplementary Fig. 1a). Doxycycline was added following infection (1 μg/ml for 48 h) and flow cytometry was used to sort cells that expressed both mCherry and BFP. These cells were then grown in the absence of doxycycline for 14 days until mCherry fluorescence returned to uninduced levels. For enCRISPRi or enCRISPRa experiments, the cells were transduced with lentiviruses containing sequence-specific sgRNAs or nontargeting sgGal4 with zsGreen1 selection marker. The transduced cells were FACS sorted for zsGreen1 positive population, induced with Dox for 72 h to activate dCas9 expression, and processed for downstream analyses such as qRT-PCR and ChIP-seq. Transient transfections were performed in 24-well plates using 500 ng of dCas9 expression vector and 250 ng of equimolar pooled or individual sgRNA expression vectors mixed with FuGENE® 6 (Promega) following the manufacturer's instructions. The transduced cells were treated with Dox for 72 h to activate dCas9 expression, and processed for downstream analyses.

**Cell growth assays.** Cell proliferation was determined using the PrestoBlue Cell Viability Reagent (Invitrogen). 5000 cells/well were seeded in triplicate into 96-well plates. After various days of culture, 10 μl of PrestoBlue reagents were added to wells with cells or medium (blank), relative absorption values at 570 and 600 nm were read after 1 h incubation at 37 °C.

**RNA isolation and qRT-PCR analysis.** RNA was isolated using RNeasy Plus Mini Kit (Qiagen) following the manufacturer's protocols. For qRT-PCR, RNA was reverse-transcribed using iScript cDNA Synthesis Kit (Bio-Rad) following the manufacturer's protocols. Quantitative RT-PCR (qRT-PCR) was performed in duplicate with the iQ SYBR Green Supermix (Bio-Rad) using CFX384 Touch Real-Time PCR Detection System (Bio-Rad). PCR amplification parameters were 95 °C

(3 min) and 45 cycles of 95 °C (15 s), 60 °C (30 s), and 72 °C (30 s). Primer sequences are listed in Supplementary Table 2.

**Western blot analysis**. Western blot was performed as described[62] using antibodies against TAL1 (Santa Cruz Biotechnology, sc-12984), β-tubulin (Cell Signaling, 2128), HA (Santa Cruz Biotechnology, sc-805), and Cas9 (Abcam, ab191468) with 1:1000 dilutions. Briefly, whole cell lysates were prepared, separated on a SDS-PAGE gel, and transferred to Amersham™ Hybond™ P 0.45 PVDF blots (GE Healthcare #10600023). The blots were incubated with primary antibodies with 5% non-fat milk in TBS/T (20 mM Tris-HCl, pH 7.5, 150 mM NaCl, 0.1% Tween-20) at 4 °C overnight with shaking. After washing 3 times with TBS/T, the blots were incubated with secondary antibodies with 5% non-fat milk in TBS/T for 1 h at room temperature. The blots were then washed 3 times with TBS/T and developed using Plus-ECL (PerkinElmer #NEL104001EA). Densitometry quantification was performed using ImageJ software.

**Phenotypic analysis of hematopoiesis**. Blood was collected via the retro-orbital plexus and complete blood counts (CBC) were performed on a HEMAVET HV950 (Drew Scientific) according to the manufacturer's protocol. Cytospin preparations were stained with May–Grunwald–Giemsa as described previously[63]. The hematopoietic stem/progenitor cell populations in mouse bone marrow were analyzed as previously described[61], including HSC (Lin−Sca1+Kit+CD150+CD48−), MPP (Lin−Sca1+Kit+CD150−CD48−), LSK (Lin−Sca1+Kit+), CMP (Lin−Sca1−Kit+CD34+CD16/32−), GMP (Lin−Sca1−Kit+CD34+CD16/32+), and MEP (Lin−Sca1−Kit+CD34−CD16/32−). Briefly, BM cells were obtained by crushing femurs, tibias, vertebrae and pelvic bones with a mortar in Ca²⁺ and Mg²⁺-free Hank's buffered salt solution (HBSS, Gibco) supplemented with 2% heat-inactivated bovine serum (HIBS, Gibco). Spleens were dissociated by crushing followed by trituration. All BM and spleen cell suspensions were filtered through 70 μm cell strainers, followed by cell counting using a Vi-CELL cell viability analyser (Beckman Coulter). For flow cytometric analysis, cells were incubated with combinations of fluorophore-conjugated antibodies. Lineage markers for HSCs and progenitors were CD2, CD3, CD5, CD8, B220, Gr1 and Ter119. Antibody staining was performed at 4 °C for 30 min or on ice for 90 min. Biotinylated antibodies were visualized by incubation with PE/Cy7-conjugated streptavidin at 4 °C for 30 min. DAPI (4,6-diamidino-2-phenylindole; 2 μg/ml in PBS) was used to exclude dead cells. Analysis or sorting was performed using a FACSAria or FACSCanto flow cytometer (BD Biosciences). Data were analyzed using FACSDiva (BD Biosciences). The following antibodies were used for flow cytometry: PerCP/Cy5.5 anti-mouse B220 (Biolegend, Cat# 103236), PE anti-mouse CD150 (Biolegend, Cat# 115903), BV510 anti-mouse CD16/32 (Biolegend, Cat# 101333), Alexa Fluor 700 anti-mouse CD3 (Biolegend, Cat# 100216), Biotin anti-mouse CD34 (eBioscience, Cat# 13-0341-85), PE anti-mouse CD43 (eBioscience, Cat# 12-0431-83), Alexa Fluor 700 anti-mouse CD48 (Biolegend, Cat# 103426), FITC anti-mouse CD71 (BD Biosciences, Cat# 553266), APC anti-mouse c-Kit (Biolegend, Cat# 105811), PE/Cy7 anti-mouse Gr1 (Biolegend, Cat# 108415), APC anti-mouse IgM (Biolegend, Cat# 406509), APC-eFluor 780 anti-mouse Mac1 (eBioscience, Cat# 47-0112-82), PerCP/Cy5.5 anti-mouse Sca-1 (Biolegend, Cat# 108123), BV510 anti-mouse Ter119 (Biolegend, Cat# 116237), FITC anti-mouse CD2 (eBioscience, 11-0021-81), FITC anti-mouse CD3 (Biolegend, Cat# 100204), FITC anti-mouse CD5 (Biolegend, Cat# 100606), FITC anti-mouse CD8a (Biolegend, Cat# 100706), FITC anti-mouse B220 (eBioscience, 11-0452-85), FITC anti-mouse Gr-1 (Biolegend, Cat# 108406), FITC anti-mouse Ter119 (Biolegend, Cat# 116206), PE/Cy7 Streptavidin (Biolegend, Cat# 405206). All antibodies were used at 1:200 to 1:400 dilutions following manufacturers's instructions.

**RNA-seq and data analysis**. RNA-seq library was prepared using the Ovation RNA-seq system (NuGEN). Sequencing reads from all RNA-seq experiments were aligned to human (hg19) reference genome by STAR 2.5.2b[64] with the parameters: --outFilterMultimapNmax 1. RSEM was used to calculate normalized gene expression (Transcripts per Million Reads or TPM)[65]. Differential gene expression analysis was performed by DESeq2[66].

**ChIP-seq and data analysis**. ChIP-seq was performed as described[34] using antibodies for HA (Santa Cruz Biotechnology, sc-805), Cas9 (Abcam, ab191468), H3K4me1 (Abcam, ab8895), H3K4me2 (Millipore, 07-030), H3K27ac (Abcam, ab4729), H3K9me3 (Abcam, ab8898), H3K27me3 (Millipore, 07-449), GATA1 (Abcam, ab11852), TAL1 (Santa Cruz Biotechnology, sc-12984), and CTCF (Millipore, 07-729) in HEK293T and/or K562 cells, respectively. Briefly, cross-linked chromatin was sonicated in RIPA 0 buffer (10 mM Tris-HCl, 1 mM EDTA, 0.1% sodium deoxycholate, 0.1% SDS, 1% Triton X-100, 0.25% Sarkosyl, pH8.0) to 200–500 bp. Final concentration 150 mM NaCl was added to the chromatin and antibody mixture before incubation overnight at 4 °C. Chromatin was washed and ChIP DNA was purified. ChIP-seq libraries were generated using NEBNext Ultra II DNA library prep kit following the manufacturer's protocol (NEB), and sequenced on an Illumina NextSeq500 system using the 75 bp high-output sequencing kit. ChIP-seq raw reads were aligned to the human hg19 genome assembly using Bowtie2[67] with the default parameters. Only tags that uniquely mapped to the

genome were used for further analysis. ChIP-seq peaks were called by MACS using the "--nomodel" parameter[68]. Peaks that overlap with the ENCODE blacklist regions[3] or the validated nontargeting sgRNA (sgGal4) enriched regions (chr6:74,229,700−74,231,800; chr3:17,443,100−17,444,200 and chr15:68,131,900−68,133,000) were removed. To compare ChIP-seq signal intensities in samples prepared from cells expressing the target-specific sgRNAs versus the nontargeting sgGal4, MAnorm[69] was applied to remove systematic bias between samples and then calculate the normalized ChIP-seq read densities of each peak for all samples. The window size was 1000 bp, which matched the average width of the identified ChIP-seq peaks.

**In vivo enhancer perturbation screen and data analysis**. CD45.2+ BM lineage negative HSPCs from dCas9-KRAB KI mice were isolated by LS columns (Miltenyi Biotec) and cultured in S-clone SF-O3 medium (Iwai North America) containing 1% FBS, 1%P/S, 1% supplement, 50 μM β-Mercaptoethanol, 50 ng/ml mouse SCF and 50 ng/ml mouse TPO. After overnight culture, cells were transduced with retroviruses containing target-specific or nontargeting control sgRNAs at MOI ≤ 0.5. Twenty million cells per mouse were transplanted into CD45.1+ lethally irradiated C57BL/6 recipient mice, followed by Dox (2 mg/ml, supplemented with sucrose at 10 mg/ml) administration in drinking water to induce dCas9-KRAB expression for 16 weeks. By this time, all donor-derived hematopoietic cells (CD45.2+) were differentiated from transplanted donor HSCs instead of short-lived progenitors. We collected cells before bone marrow transplantation (T1) as the baseline control and the GFP-positive donor-derived BM HSPCs (Lin−Sca1+Kit+) and mature myeloid (Gr1+Mac1+), B (B220+) and T (CD3+) lymphoid cells in the peripheral blood of recipients 16 weeks post transplantation (T2). Genomic DNA was isolated and sgRNA sequences were PCR amplified using primers listed in Supplementary Table 2. PCR amplicon libraries were generated using NEBNext Ultra II DNA library prep kit following the manufacturer's protocol (NEB), and sequenced on an Illumina NextSeq500 system using the 75 bp high-output sequencing kit. Two biological replicate experiments for each enCRISPRi screen were performed. In each replicate experiment, myeloid, B and T cells were isolated from 3 (for locus-specific perturbation) or 15 (for multiloci perturbation) independent recipient mice, and genomic DNA was isolated and pooled for the sgRNA amplicon sequencing analysis. After sequencing, the sgRNA sequences were extracted from raw fastq files and mapped to sgRNA sequences in enCRISPRi screen libraries. Reads of each sgRNA were counted and normalized to the total read counts of each sample. Mean sgRNA counts from replicates were calculated at starting (T1) and end time point (T2). The growth phenotype of each sgRNA was quantified as log2 transformed mean sgRNA count ratio between T2 and T1. For calculation of the significance (false discovery rate or FDR) of depletion for each targeted enhancer, promoter or nontargeting control (NC) regions in each cell type, we used MAGECK[70] test with the parameters of --norm-method total --gene-lfc-method mean.

**Statistics and reproducibility**. Statistical details including N, mean and statistical significance values are indicated in the text, figure legends, or methods. Error bars represent standard error of the mean (SEM) or standard deviation (SD) from either independent experiments or independent samples. All statistical analyses were performed using GraphPad Prism, and the detailed information about statistical methods is specified in figure legends or methods. The numbers of independent experiments or biological replicate samples and P values (n.s. not significant, *P < 0.05, **P < 0.01, ***P < 0.001) are provided in individual figures. P < 0.05 was considered statistically significant. Panels in Fig. 4d, g, l, 5b, c, and Supplementary Figs. 6d show a representative image or flow cytometry of at least three independent experiments or biological replicate samples. Analysis of gene expression was determined using the $2^{-\Delta\Delta Ct}$ method with *GAPDH* as the reference gene unless otherwise specified. No statistical method was used to predetermine sample size in animal studies and the experiments were not randomized. The investigators were not blinded to allocation during experiments and outcome assessment.

**Reporting summary**. Further information on research design is available in the Nature Research Reporting Summary linked to this article.

## Data availability

All raw and processed ATAC-seq, ChIP-seq and RNA-seq data are available in the Gene Expression Omnibus (GEO): GSE132216. The source data underlying Figs. 1b–e, 2b, f, 4b–h, j, k, and 5c, and Supplementary Figs. 1b–d, f, h, 5a–h, 6a–c, e, f, 10a–e, and 11a–d are provided as a Source Data file. The plasmids used in this study are available from Addgene (#138456, #138457, #138458, #138459, #138460, #138461, #138462 and #138463). All other data supporting the findings of this study are available from the corresponding author on request.

## Code availability

The computer code for sgRNA counting is available from GitHub (https://github.com/Yuxuan0060/CRISPR_screen). Other codes are available from the corresponding author on request.

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

## Acknowledgements

We thank the BioHPC computational infrastructure at UTSW for assistance, Lin Li and Le Qi at Children's Research Institute (CRI) at UTSW for technical support, and Luke A. Gilbert, Stanley S. Qi and Jonathan S. Weissman at UCSF for providing the CRISPRi constructs. K.L. and Y.L. were supported by the Cancer Prevention and Research Institute of Texas (CPRIT) training grant (RP160157). X.L. was supported by the American Heart Association postdoctoral fellowship (18POST34060219). J.X. is a Scholar of The Leukemia & Lymphoma Society. This work was supported by the NIH grants R01DK111430 and R01CA230631, the CPRIT grants (RR140025, RP180504, RP180826 and RP190417), and the Welch Foundation grant I-1942 (to J.X.).

## Author contributions

Conceptualization, K.L., Y.L. and J.X.; Methodology, K.L., Y.L., H.C., Y.Z., Z.G., X.L., A.Y., P.K., K.E.D., M.N. and J.X.; Investigation, K.L., X.L., H.C., P.K. and J.X.; Writing—original draft, K.L., Y.L. and J.X.; Writing—review and editing, J.X.; Funding acquisition, J.X.; Supervision, J.X.

## Competing interests

The authors declare no competing interests.
