## [Peer Review File · Nature Communications]

Reviewers' Comments:

Reviewer #1:

Remarks to the Author:

Functional evaluation of cis-regulatory elements is challenging. To address this issue, Xu and colleagues have developed improved CRISPR/dCas9-effector systems that allow for synergistic action of effectors at target cis-regulatory elements. For gene activation from cis-regulatory elements, they established and tested dCas9 with a combination of p300 and VP64 effectors. For cis-regulatory element perturbation, they established a combinatorial KRAB and LSD1 effector strategy. These synergistic systems are tested in comparison with other available effectors and single-effector strategies, and evaluation at multiple loci occurs in cell culture systems. In addition, xenotransplantation and bone marrow transfer experiments are used to describe use of the system in an in vivo setting. The latter includes description of a novel knock-in mouse model for inducible dCas9-KRAB expression.

Overall, quality of the manuscript and its data is high, and next generation sequencing based readouts provide a nice overview on effector function. In addition, the evaluation of the dCas9-effector approach in different model systems and loci strengthen the manuscript. The data are technically sound and support overall conclusions.

While effector and strategies used in the study are not novel, the synergistic system and the development of a mouse model significantly adds to the CRISPR toolbox and is expected to be adapted by many investigators beyond the specialist field of gene regulation.

Overall, I think the manuscript is of sufficient content and quality to warrant publication in Nature Communications.

Minor comment:

With the experiment as presented, the authors cannot rule out that steric hindrance could be at least partly mediating the repressive effects they see. Indeed, proximity to the ATAC-seq peaks benefits repressive effects. While the authors discuss this observation in general, steric hindrance is not mentioned. This caveat should be discussed.

Reviewer #2:

Remarks to the Author:

In this study, the authors developed a dCas9 based double effector fusion CRISPRa or CRISPRi system using the MS2 scaffold. The authors fused p300-VP64 and showed that it improves the CRISPRa system in K562 cells targeting amongst other the enhancers of b-globin locus. They compared different effectors VPR, SunTag, SAM with their enCRISPRa system and found it to be better in upregulating b-globin genes and MYOD, but quite comparable to other effector systems for IL1RN and OCT4 genes in K562 cells. Then they developed enCRISPRi system using LSD1-KRAB (LK) or KRAB-LSD1(KL) fusion. They found both of these fusions to be efficient at downregulating their target genes. They ran an impressive 40 ChIP assays to assess the chromatin marks on the targeting locus in K562 cells. Furthermore, they used enCRISPRa system in jurkat cells where there is a heterozygous 12bp insertion in TAL1 enhancer in jurkat cells. They designed two guides that harbour PAM sequence within 12bp insertion (sgMut1,2) and two guides targeting wild type allele sgWT1,2. They found sgMut2 and sgWT2 to be better at upregulating TAL1 in jurkat cells. They then xenografted these in NOD-scidIL2Rgnull mice to show that higher expression of TAL1 leads to higher tumor burden. Finally, they used enCRISPRi ex-vivo to downregulate hematopoietic lineage specific enhancers. Their hypothesis is "If enCRISPRi-mediated repression of an enhancer or promoter impaired its function and target gene expression, the corresponding sgRNAs would be depleted (or enriched) in T2 relative to T1 cells". The authors throughout measured the sgRNA abundance or depletion to draw conclusions. The authors have done a nice job in some places, but

some results sections need more work to make this article fitting for publication in Nature Communications and many of their claims need to be toned down as the authors do not have evidence in the article to support them. I have the following comments:

Major

- 'These data emphasize the importance of targeting dCas9-based epigenetic editing complexes to the most accessible regions at enhancers for the maximal transcriptional perturbation.' You don't have the N for providing this statement, both in terms of guide coverage and more importantly number of enhancers. To really test this you would need a more comprehensive analyses of thousands of guides and enhancers. I would tone this very much down both here and in discussion and mention caveats.

TAL1: The TAL1 results section needs more work.

Off-target: The authors did not look at other than green highlighted PAM sequences available in WT and Mut alleles? And if the sgMut1 and 2 can target other sites in WT allele as well, or vice versa, sgWT2 can target Mut allele since the sgRNA targeting is known to tolerate several mismatches.

enCRISPRi: More importantly this result show that upregulation of TAL1 expression leads to tumor burden, but doesn't prove allele specific targeting of TAL1. In fact can authors show that enCRISPRi using same guides can lead to reduction of TAL1 and hence reduction in tumor burden.

Allele specific upregulation: Is there any way authors can differentiate between the TAL1 cDNA from one allele to the other to show allele specific upregulation?

Mouse enCRISPRi:

Enhancer sgRNAs not repressing: The authors could see downregulation of Runx1 gene by targeting its promoter but not its enhancers and they concluded that "enCRISPRi-mediated enhancer repression was ineffectual, supporting the non-essential roles of Runx1 enhancers in myelopoiesis or lymphopoiesis". Especially when they tested just 2 guide RNA per enhancer and both of them did not show any downregulation of Runx1 (FigS9). Either these are not enhancers of Runx1, or in the cells they tested or they have not saturated enough sgRNA targeting to draw this conclusion. They could have tested that these targeting guides can lead to downregulation in certain cells (cells where they were identified as enhancers) to provide evidence for the sgRNA targeting efficiency. Similar is the case with GATA1 or GATA2 enhancers.

CEBPA: For CEBPA authors concluded that, "E1 or E3 are indispensable" based on inefficiency of downregulation observed and lack of sgRNA depletion. I think the authors cannot draw this conclusion in the absence of strong evidence. More importantly, authors did not show any neighboring gene expression data to confirm if the enhancer is specific to the target gene.

Minor

- 'CRSIPR' misspelled in abstract.

- In ""Enhancers are cis-regulatory DNA sequences that are bound and regulated by transcription factors (TFs) and chromatin regulators in a highly tissue-specific manner." Enhancers are also temporal specific and even cell-type specific within a tissue.

- 'however, the in vivo functions of the vast majority of these elements within their native chromatin remain unknown.' And later 'Additionally, conventional gene targeting or genome editing approaches have been utilized to knockout (KO) or mutate specific enhancers in cell lines or animal models'. Disagree with these statements. There are many large-scale CRISPR screens

now available, both regular editing and CRISPRi/a that have tested the function of many elements in cells and also identify their target gene. I would tone this down and potentially cite some of these or a review of them.

-'Furthermore, high-resolution saturating screens of cis-regulatory elements rely on loss-of-function and do not permit gain-of-function analyses'. Disagree also. These could easily be done as gain of function also. Would tone down.

Would be good to provide some brief detail in intro on the various CRISPRa systems used and compared here (i.e. VPR, SunTag, SAM).

-'Here we develop' comma after 'Here'.

-'Notably, enCRISPRa-mediated HS2enhancer activation led to 13.0 to 40.6-fold increases in expression of β -globin genes HBE1, HBG1/2 and HBB relative to the non-transduced controls, which are significantly higher than other dCas9 activation methods (Fig. 1c)'. Rephrase to make it clear enhancer activation led to a 13, 24 and 40.6-fold increase in genes respectively.

-In the stable line work: The Authors mentioned sorting according to mCherry and BFP while no indication of where the BFP signal is coming from (my guess is the rTA). Also, what measures were done to make sure that stable cell line indeed included the dCas9 construct after FP marks went down to uninduced levels and what copy numbers were similar across the lines - should have a supplementary figure on that for all constructs.

-The last paragraph of the section 'Generation of an Inducible Knock-In Mouse Model for In Vivo enCRISPRi' feels more like the beginning of the other section. Would probably move it there.

-'while all four Cebpa enhancers share similar chromatin accessibility and H3K27ac enrichment in HSCs and/or myeloid cells, only +8kb E2 and +37kb E4 enhancers are indispensable for myeloid development' again too bold a statement and would tone down. You have not checked all possible chromatin marks for these.

- TAL1 and GATA1 effect seen - would you expect that just by targeting dCas9 to the genomic location? Do the author postulate that the epigenetic editing affect TF binding? Would be great to discuss this more.

- For enCRISPRi the authors produced and tested both possible combinations of the dual effectors (indicated as LK and KL). Would be interesting to discuss what would happen with the positive effectors if more than one was tested together.

-In the statement "but also identify selective 'vulnerabilities' of enhancers that may be employed to precisely control gene expression". What are they kind of "vulnerabilities" the authors meant?

-You mention therapeutics briefly in discussion but while possibly OK for enCRISPRi, might be too high levels for some things in terms of enCRISPRa and can be dangerous for the patient. Also, don't think these constructs could fit into the commonly used therapeutic AAV vector. All of this should be mentioned and discussed.

-Figure 1B and C - lacking in figure or description n=?

-Figure 1C - distance scales will be helpful

-Figure 1D - aside from the on- and off-target loci, there are a couple of loci coming up with over a 100 counts. It would be interesting to elaborate on those, rather they are in the HBG locus or outside of it, and rather those are contact point to other genes. This information is crucial for later in vivo tests.

-Figure 2 – The authors claim that HBB and HBG1 are affected by the enCRISPRi. If so, then it seems that many other genes are affected as well. Maybe another representation of the data is better where both log₂FC and the pValue is plotted?

-Figure 2E - please check the scales of the tracks indicated in the zoomed and expanded images. For example the DHS in the zoomed is 0-300 while in the expanded track is 0-80 and in both it seems to reach the maximum but doesn't exceed it.

-In Figure 3 enCRISPRi(LK) – (C/K/L/KL) tracks. Please write a description either in the figure legend or in the Methods

-Figure 5D – use either Spi1/PU.1 or PU.1 in text and figure at all places

Reviewer #3:

Remarks to the Author:

In this manuscript, Li et al. describe the development and testing of enhanced systems for CRISPRi and CRISPRa with the goal of modulating enhancer activity. The method combines dCas9 tagging with the MS2-MCP system for recruiting additional regulatory domains. They show efficacy at a handful of enhancers in vitro and perform a small screen in vivo. The authors also analyze the effects on histone modifications and TF binding at the targeted loci. While the study is overall well designed, we don't believe that it warrants publication in Nature Communications as this is a very small advance in the context of CRISPR-based epigenome editing.

1. The authors describe a "new" CRISPR-based epigenome engineering approach that they term enCRISPRi and enCRISPRa, which involves dCas9 fused to regulatory domains with additional MCP fused regulatory domains recruited via MS2 loops on guide RNAs. This strategy was published by Konermann et al in 2015 and named SAM. While the authors have changed the regulatory domains (to other previously published domains), the concept is exactly the same and therefore represents a minor advance. In addition, there are several papers that describe the use of multiple regulatory domains at the same time (Konermann et al Nature 2015, Chavez et al Nature Methods 2015, Carleton et al Cell Systems 2017, and Yeo et al Nature Methods 2018) and multiplex regulation, the regulation of multiple genes simultaneously, has been shown by at least three groups (Konermann et al Nature 2015, Carleton et al Cell Systems 2017, and Yeo et al Nature Methods 2018). While the in vivo section adds some novelty, the experiment is more similar to an ex vivo study (where the epigenetic engineering is occurring) with in vivo phenotyping.

2. Related to the point above, the manuscript is written in such a way that makes the developed method appear completely unique. The introduction and conclusion set the tone that these enCRISPR methods are special in their capabilities, but several studies have used variants of CRISPRa and CRISPRi successfully at enhancers (e.g. Hilton et al 2014, Klann et al, Carleton et al Cell Systems 2017).

3. The term "enhanced crispr", which is used in the title and name of the method, has already been used and should be changed to avoid confusion.

4. Guide RNAs without MS2 loops should be used as a control. In addition, we could not find a direct comparison of enCRISPRa to P300 and VP64 alone.

5. In the introduction, the authors state that the effect of dCas9 fusions declines rapidly when the target regions move away from the proximal promoter. But references 18-20 don't really show this.

6. The authors state that the same guide RNAs were used in each experiment shown in Figure 1. Do these guide RNAs all include MS2 loops?
7. The authors state that H3K27me3 was not observed at the beta-globin locus with or without CRISPRi techniques, but the data is not shown. The data should be shown or the statement should be left out.
8. In several places in the manuscript (bottom of page 7, 2nd paragraph of page 8, and 1st paragraph of the discussion), the authors state that there is cooperation between repressor domains, but this hasn't been shown. The additional repression observed using enCRISPRi could be due to multiple copies of the MCP fused receptor and not cooperation or synergy. The authors should use an unfused dCas9 with MCP fusions as a comparison.
9. The authors state that enCRISPRi effects are constrained by CTCF, since CTCF binding is not affected. However, the authors do not have any evidence that CTCF has anything to do with confinement. The authors should simply state that CTCF binding is unaffected by enCRISPRi.
10. ChIP-seq signal at control regions that were not targeted should be shown (possibly as a supplemental figure) to assure comparable signal between ChIP-seq data sets overall.
11. In the section on allele-specific activation of TAL1, the results look promising but allele-specificity hasn't been proven. Allele-specific targeting should be shown by ChIP and a control of a wildtype cell not responding to the mutant guide RNAs should be shown.
12. In Figure 4E, why is there cell luminescence observed at 4 hours post implantation? If there is luminescence 4 hours post implantation, why is there no luminescence at 0 weeks post-transplant in Figure 4F?
13. In the screens in the last two sections of the results, the authors often interpret no effect on phenotype as the enhancer not playing a functional role. It is very difficult to interpret negative results from this type of experiment. If the authors want to conclude negative results (e.g. that E1 and E3 of Cebpa are dispensable), then efficient targeting of these sites needs to be shown to rule out the simple explanation that the guide RNAs targeting the dispensable sites simply do not work.
14. As discussed in response 1 above, the authors state that they are performing multiplex perturbations; however, the field uses the word multiplex as the simultaneous (i.e. in the same cells) perturbation of multiple genes. This type of experiment has not been performed in this study, instead I would refer to the section on multiplex perturbations as a small screen to avoid confusion.
15. Are the p-values in Figure S6 adjusted for multiple hypothesis testing? If not, the p-values should be corrected.
16. In the first paragraph of the discussion, the authors state that epigenetic writer proteins specifically modulate histone modifications. However, VP16 and KRAB are fairly non-specific chromatin modifiers that recruit several different cofactors.
17. In the data shown in Figures 1-3, what is the timing of the experiment, with respect to Dox induction, for assaying expression or protein:DNA interactions?
18. In all bar graphs, every point should be shown as well (as was done in Figure 4B).

Summary of Revisions

We are grateful to the reviewers and editors for the thoughtful feedback on our manuscript. The reviewers have commented that “*the manuscript is of sufficient content and quality to warrant publication in Nature Communications*” (Reviewer 1), “*The authors have done a nice job in some places, but some results sections need more work to make this article fitting for publication in Nature Communications*” (Reviewer 2), and “*the study is overall well designed*” (Reviewer 3). However, the reviewers also raised questions related to the tone of some statements, additional control experiments and validation studies to make the study more convincing.

Following the reviewers’ suggestions and editorial instructions, we have performed additional experiments, data analysis and textual revisions to address the points raised by the reviewers.

The major changes are detailed below:

- 1) We included the comparisons between the dual-effector enCRISPRa and individual dCas9-effectors (Fig. 1b,c), as suggested by Reviewers 2 and 3.
- 2) We included additional control experiments or analyses including sgRNAs without MS2 loops, unfused dCas9, H3K27me3 ChIP-seq, and ChIP-seq profiles at non-targeted genomic loci (Fig. 3; Supplementary Figs. 1c,d, 2 and 4), as suggested by Reviewer 3.
- 3) We included results showing allele-specific binding of dCas9 by mutant allele-targeting sgRNAs at the *TAL1* oncogenic super-enhancer by ChIP experiments (Fig. 4b), as suggested by Reviewers 2 and 3.
- 4) As suggested by Reviewer 2, we included parallel analyses of *TAL1* oncogenic super-enhancer using enCRISPRi, and found that enCRISPRi-mediated repression of *TAL1* enhancer downregulated *TAL1* expression and impaired T-ALL cell growth (Fig. 4f-h).
- 5) We included experiments to determine the targeting efficiency of dCas9 complexes to various enhancers and promoters by ChIP experiments in primary HSPCs (Supplementary Fig. 11a-d). We also included experiments to show that enCRISPRi-mediated repression impaired the expression of enhancer-associated genes (e.g. *Spi1*, *Runx1*, *Gata1*, *Gata2* and *Cebpa*) but not the other neighboring genes (Supplementary Fig. 10a-e), as suggested by Reviewers 2 and 3.
- 6) We included a diagram for all the constructs used in this study (Supplementary Fig. 1a), and the analyses of the top dCas9 off-target loci (Supplementary Fig. 1e-h), as commented by Reviewers 2 or 3.
- 7) We significantly revised the texts including introduction, conclusion and discussion to better describe the existing literature/knowledge, and the technical and conceptual advances that our work present in light of previous publications, as commented by Reviewer 3.

In our view, these new results not only addressed the reviewers’ questions, but also extended our previous findings by providing more definitive evidence to establish the enCRISPRi and enCRISPRa systems in enhancer perturbation *in vitro*, in xenografts, and *in vivo*.

A point-by-point response to the reviewers’ comments is provided below. The reviewers also had a number of comments regarding technical details, clarifications and discussion points that we believe to have been adequately addressed with additional experiments, data analysis, and/or textual revisions.

Here we provide our point-by-point response (in *blue*) to the reviewers' comments.

Reviewer #1 (Remarks to the Author):

Functional evaluation of cis-regulatory elements is challenging. To address this issue, Xu and colleagues have developed improved CRISPR/dCas9-effector systems that allow for synergistic action of effectors at target cis-regulatory elements. For gene activation from cis-regulatory elements, they established and tested dCas9 with a combination of p300 and VP64 effectors. For cis-regulatory element perturbation, they established a combinatorial KRAB and LSD1 effector strategy. These synergistic systems are tested in comparison with other available effectors and single-effector strategies, and evaluation at multiple loci occurs in cell culture systems. In addition, xenotransplantation and bone marrow transfer experiments are used to describe use of the system in vivo setting. The latter includes description of a novel knock-in mouse model for inducible dCas9-KRAB expression.

Overall, quality of the manuscript and its data is high, and next generation sequencing based readouts provide a nice overview on effector function. In addition, the evaluation of the dCas9-effector approach in different model systems and loci strengthen the manuscript. The data are technically sound and support overall conclusions.

While effector and strategies used in the study are not novel, the synergistic system and the development of a mouse model significantly adds to the CRISPR toolbox and is expected to be adapted by many investigators beyond the specialist field of gene regulation.

Overall, I think the manuscript is of sufficient content and quality to warrant publication in Nature Communications.

We appreciate the positive feedback on our manuscript.

Minor comment:

With the experiment as presented, the authors cannot rule out that steric hindrance could be at least partly mediating the repressive effects they see. Indeed, proximity to the ATAC-seq peaks benefits repressive effects. While the authors discuss this observation in general, steric hindrance is not mentioned. This caveat should be discussed.

We thank the reviewer for the insightful comment. As suggested, we included in the Discussion under "Considerations for Enhancer Perturbations by CRISPR Epigenetic Editing" that steric hindrance of dCas9 binding should be considered when designing enCRISPRi/a assays. Although we designed sgRNAs to avoid overlapping with known transcription factor (TF) binding sites (as indicated by ATAC-seq signals), it is possible that the binding of the dCas9-effector complexes in the proximity of TF binding sites may interfere with the assembly of multiple-TF complexes on chromatin due to steric hindrance.

Reviewer #2 (Remarks to the Author):

In this study, the authors developed a dCas9 based double effector fusion CRISPRa or CRISPRi system using the MS2 scaffold. The authors fused p300-VP64 and showed that it improves the CRISPRa system in K562 cells targeting amongst other the enhancers of b-globin locus. They compared different effectors VPR, SunTag, SAM with their enCRISPRa system and found it to be better in upregulating b-globin genes and MYOD, but quite comparable to other effector systems for IL1RN and OCT4 genes in K562 cells. Then they developed enCRISPRi system using LSD1-KRAB (LK) or KRAB-LSD1(KL) fusion. They found both of these fusions to be efficient at downregulating their target genes. They ran an impressive 40 ChIP assays to assess the chromatin marks on the targeting locus in K562 cells. Furthermore, they used enCRISPRa system in jurkat cells where there is a heterozygous 12bp insertion in TAL1 enhancer in jurkat cells. They designed two guides that harbour PAM sequence within 12bp insertion (sgMut1,2) and two guides targeting wild type allele sgWT1,2. They found sgMut2 and sgWT2 to be better at upregulating TAL1 in jurkat cells. They then xenografted these in NOD-scidIL2Rnull mice to show that higher expression of TAL1 leads to higher tumor burden. Finally, they used enCRISPRi ex-vivo to downregulate hematopoietic lineage specific enhancers. Their hypothesis is "If enCRISPRi-mediated repression of an enhancer or promoter impaired its function and target gene expression, the corresponding sgRNAs would be depleted (or enriched) in T2 relative to T1 cells". The authors throughout measured the sgRNA abundance or depletion to draw conclusions. The authors have done a nice job in some places, but some results sections need more work to make this article fitting for publication in Nature Communications and many of their claims need to be toned down as the authors do not have evidence in the article to support them. I have the following comments:

Major

-‘These data emphasize the importance of targeting dCas9-based epigenetic editing complexes to the most accessible regions at enhancers for the maximal transcriptional perturbation.’ You don’t have the N for providing this statement, both in terms of guide coverage and more importantly number of enhancers. To really test this you would need a more comprehensive analyses of thousands of guides and enhancers. I would tone this very much down both here and in discussion and mention caveats.

We appreciate this thoughtful comment, and agree with the reviewer that more Ns would be needed for a strong statement. As suggested, we have removed this statement in the results and mentioned caveats in the discussion.

TAL1: The TAL1 results section needs more work.

Off-target: The authors did not look at other than green highlighted PAM sequences available in WT and Mut alleles? And if the sgMut1 and 2 can target other sites in WT allele as well, or vice versa, sgWT2 can target Mut allele since the sgRNA targeting is known to tolerate several mismatches.

enCRISPRi: More importantly this result show that upregulation of TAL1 expression leads to tumor burden, but doesn’t prove allele specific targeting of TAL1. In fact can authors show that enCRISPRi using same guides can lead to reduction of TAL1 and hence reduction in tumor burden.

Allele specific upregulation: Is there any way authors can differentiate between the TAL1 cDNA from one allele to the other to show allele specific upregulation?

We appreciate these thoughtful comments. It is important to note that the sgRNAs for the WT allele (sgWT1 and sgWT2) were used as controls in these experiments. Specifically, sgWT1 should target enCRISPRa to both alleles since the sgRNA target sequence is shared between WT and mutant (Mut) alleles. Similarly for sgWT2 since the first 14 nucleotides upstream of the PAM site are shared between WT and Mut alleles.

To directly validate the allele-specific binding of enCRISPRa by Mut allele-specific sgRNAs (sgMut1 and sgMut2), we performed ChIP experiments using the antibody against dCas9 in Jurkat and K562 cells, respectively (Fig. 4b). We first designed PCR primers that distinguish WT and Mut alleles based on the 12bp insertion sequence at the *TAL1* oncogenic super-enhancer (SE) (Supplementary Table 2). In Jurkat cells that carry both WT and Mut alleles, expression of sgWT1 or sgWT2 with enCRISPRa led to significant and comparable dCas9 binding at both alleles compared to the non-targeting sgGal4 control. By contrast, sgMut1 or sgMut2 resulted in significant dCas9 binding at the Mut but not the WT allele (Fig. 4b). In K562 cells that contain only WT alleles, expression of sgWT1 or sgWT2 led to significant dCas9 binding at the WT allele, whereas no significant dCas9 binding at either allele was noted upon the expression of sgMut1 or sgMut2. These new results provide direct evidence that sgMut1 or sgMut2 targets the dCas9 complexes specifically to the Mut allele, whereas sgWT1 or sgWT2 targets dCas9 to both WT and Mut alleles. We agree with the reviewer that it would be interesting to determine the allele-specific upregulation of *TAL1*. Such analyses require the knowledge of the haplotype of the *TAL1* locus and *TAL1* coding SNPs that are in linkage disequilibrium with *TAL1* enhancer sequences in Jurkat cells. Since the *TAL1* super-enhancer locates ~8kb upstream of its first exon, we attempted to determine *TAL1* haplotype by long-range genomic DNA PCR without success due to technical difficulties (e.g. it required the amplification of >20kb genomic DNA fragment containing the *TAL1* enhancer sequence and ORF in a single amplicon).

As suggested, we performed additional experiments to determine whether enCRISPRi using the same guides leads to reduced *TAL1* expression (Fig. 4f-h). Specifically, we found that repression of *TAL1* SE by enCRISPRi (dCas9-LSD1 + MCP-KRAB) with sgMut1/2 or sgWT1/2 resulted in significant downregulation of *TAL1* mRNA and protein in Jurkat cells (Fig. 4f,g). Moreover, impaired *TAL1* expression by enCRISPRi led to significantly decreased cell growth (Fig. 4h). Together with the enCRISPRa results (Fig. 4c-e), these findings provide additional evidence to establish the functional role of the *TAL1* oncogenic SE in T-ALL development. The new results also demonstrate the efficacy of enCRISPRa and enCRISPRi for allele-specific activation and repression of disease-associated enhancers, respectively.

Mouse enCRISPRi:

Enhancer sgRNAs not repressing: The authors could see downregulation of Runx1 gene by targeting its promoter but not its enhancers and they concluded that “enCRISPRi-mediated enhancer repression was ineffectual, supporting the non-essential roles of Runx1 enhancers in myelopoiesis or lymphopoiesis”. Especially when they tested just 2 guide RNA per enhancer and both of them did not show any downregulation of Runx1 (FigS9). Either these are not enhancers of Runx1, or in the cells they tested or they have not saturated enough sgRNA targeting

to draw this conclusion. They could have tested that these targeting guides can lead to downregulation in certain cells (cells where they were identified as enhancers) to provide evidence for the sgRNA targeting efficiency. Similar is the case with GATA1 or GATA2 enhancers.

CEBPA: For CEBPA authors concluded that, "E1 or E3 are indispensable" based on inefficiency of downregulation observed and lack of sgRNA depletion. I think the authors cannot draw this conclusion in the absence of strong evidence. More importantly, authors did not show any neighboring gene expression data to confirm if the enhancer is specific to the target gene.

We appreciate these insightful comments. We agree with the reviewer that the lack of *Runx1* or *Cebpa* downregulation and sgRNA depletion by targeting enhancers could be due to: 1) the sgRNAs did not efficiently target enCRISPRi to the enhancer sequences; and/or 2) the targeted enhancers were not essential for *Runx1* (or *Cebpa*) expression. To distinguish between these possibilities, we have devoted extraordinary efforts to perform additional experiments as detailed below:

- (a) We determined the chromatin binding of the dCas9 complexes in primary hematopoietic stem/progenitor cells (HSPCs) isolated from the dCas9-KRAB knock-in mouse bone marrow. Upon retroviral expression of each CRE-targeting sgRNA or non-targeting sgGal4 in dCas9-KRAB expressing HSPCs, ChIP experiments using the antibody against dCas9 were performed to determine the enrichment of dCas9 binding at the sgRNA-targeted genomic loci (Supplementary Fig. 11a-d). We found that the expression of CRE-targeting sgRNAs resulted in significant enrichment of dCas9 binding at the targeted promoter or enhancer regions relative to non-targeting sgGal4. Importantly, the dCas9 targeting efficiency (as determined by the ChIP signals) was largely comparable for the tested sgRNAs at each locus, despite that only some sgRNAs resulted in significant downregulation of target genes (Supplementary Fig. 10). For example, the sgRNAs targeting *Cebpa* promoter (P-sg1) resulted in comparable dCas9 binding as the sgRNAs targeting *Cebpa* enhancers (E1-sg1, E1-sg2, E3-sg1 and E3-sg2; Supplementary Fig. 11a); however, only P-sg1 led to significant *Cebpa* downregulation (Supplementary Fig. 10a). Similar results were obtained for other tested loci including *Gata1*, *Gata2* and *Runx1* (Supplementary Fig. 11b-d). These findings suggest that the lack of target gene repression is unlikely due to inefficient dCas9/sgRNA targeting; however, it is important to note that the presence of dCas9 chromatin occupancy may not reflect the repressive function of the assembled enCRISPRi complexes on chromatin. We have also discussed this possibility in the revised manuscript.
- (b) To determine whether the enhancer is specific to the target gene, we measured the expression of the nearest neighbor genes of the targeted enhancers in primary HSPCs upon enCRISPRi-mediated repression (Supplementary Fig. 10a-e). We found that the repression of the targeted promoters or enhancers had no significant effect on the expression of other nearest neighbor genes at the tested loci. For example, while repression of the *Cebpa* promoter (by P-sg1/2) or enhancers (by E2-sg1/2 or E4-sg1/2) significantly downregulated *Cebpa* expression, no significant change in the expression of two flanking genes *Cebpg* and *Slc7a10* was noted (Supplementary Fig. 10a). These findings suggest that the targeted enhancers (E2 and E4) were specifically required for the expression of the *Cebpa* gene in HSPCs.

Minor

- 'CRSIPR' misspelled in abstract.

We corrected the typo.

- In ""Enhancers are cis-regulatory DNA sequences that are bound and regulated by transcription factors (TFs) and chromatin regulators in a highly tissue-specific manner." Enhancers are also temporal specific and even cell-type specific within a tissue.

We revised the sentence to "Enhancers are cis-regulatory DNA sequences that are bound and regulated by transcription factors (TFs) and chromatin regulators in a highly cell-type and temporal-specific manner".

- 'however, the in vivo functions of the vast majority of these elements within their native chromatin remain unknown.' And later 'Additionally, conventional gene targeting or genome editing approaches have been utilized to knockout (KO) or mutate specific enhancers in cell lines or animal models'. Disagree with these statements. There are many large-scale CRISPR screens now available, both regular editing and CRISPRi/a that have tested the function of many elements in cells and also identify their target gene. I would tone this down and potentially cite some of these or a review of them.

As suggested, we have significantly revised the introduction to tone down these statements, and cited relevant studies (pages 3-4).

-‘Furthermore, high-resolution saturating screens of cis-regulatory elements rely on loss-of-function and do not permit gain-of-function analyses’. Disagree also. These could easily be done as gain of function also. Would tone down.

We apologize for the potential misunderstanding of this statement, in which we referred to the use of Cas9-mediated saturating mutagenesis to study the function of cis-elements. Nevertheless, we revised the statement to be more specific, and also toned down the statements as suggested.

Would be good to provide some brief detail in intro on the various CRISPRa systems used and compared here (i.e. VPR, SunTag, SAM).

We added brief introductions on the first-generation (dCas9-VP64) and second-generation CRISPRa systems (SAM, SunTag and VPR) as suggested.

-‘Here we develop’ comma after ‘Here’.

We added comma as suggested.

-‘Notably, enCRISPRa-mediated HS2 enhancer activation led to 13.0 to 40.6-fold increases in expression of β -globin genes HBE1, HBG1/2 and HBB relative to the non-transduced controls, which are significantly higher than other dCas9 activation methods (Fig. 1c)’. Rephrase to make it clear enhancer activation led to a 13, 24 and 40.6-fold increase in genes respectively.

We rephrased the sentence as suggested.

-In the stable line work: The Authors mentioned sorting according to mCherry and BFP while no indication of where the BFP signal is coming from (my guess is the rTA). Also, what measures were done to make sure that stable cell line indeed included the dCas9 construct after FP marks went down to uninduced levels and what copy numbers were similar across the lines - should have a supplementary figure on that for all constructs.

We provided more details on the vector design and experimental strategy in the methods (page 22). We also included a diagram to illustrate all the constructs used in these studies (Supplementary Fig. 1a) as suggested. Briefly, the BFP signal comes from the rTA construct and mCherry from dCas9 constructs. To establish Dox-inducible dCas9-effector expressing cell lines, the parental cells were transduced with lentiviruses co-expressing rtTA-BFP and dCas9-P2A-mCherry. The cells were treated with Dox (1 μ g/mL) for 48 hours, and BFP+mCherry+ cells were FACS sorted. Cells with similar BFP and mCherry levels were sorted to ensure comparable expression of rtTA and dCas9-effectors. These cells were then cultured in the absence of Dox for 14 days until the expression of dCas9-P2A-mCherry returns to uninduced levels. For enCRISPRi or enCRISPRa experiments, the cells were transduced with lentiviruses containing sequence-specific sgRNAs or non-targeting sgGal4 with puromycin or zsGreen1 selection marker. The transduced cells were selected by puromycin or FACS sorted for zsGreen1+, induced with Dox for 72 hours to activate dCas9 expression, and processed for downstream analyses.

-The last paragraph of the section ‘Generation of an Inducible Knock-In Mouse Model for In Vivo enCRISPRi’ feels more like the beginning of the other section. Would probably move it there.

We agreed and moved the paragraph to the next section as suggested.

-‘while all four *Cebpa* enhancers share similar chromatin accessibility and H3K27ac enrichment in HSCs and/or myeloid cells, only +8kb E2 and +37kb E4 enhancers are indispensable for myeloid development’ again too bold a statement and would tone down. You have not checked all possible chromatin marks for these.

As suggested, we revised the statement to “while the annotated *Cebpa* enhancers share similar chromatin accessibility and H3K27ac enrichment in HSCs and/or myeloid cells, the enCRISPRi-mediated repression of the +8kb E2 and +37kb E4 enhancers resulted in more profound impacts on myeloid development (Figs. 5e and 6b)”.

- TAL1 and GATA1 effect seen – would you expect that just by targeting dCas9 to the genomic location? Do the author postulate that the epigenetic editing affect TF binding? Would be great to discuss this more.

We appreciate this insightful comment. We expect that just targeting dCas9 to the genomic location would not be sufficient to disrupt GATA1 and TAL1 binding in our studies because: 1) a recent report by Skotheim and colleagues described the CRISPRd method in which the binding of dCas9/sgRNA complexes can disrupt TF-DNA interactions¹. However, this method requires that the sgRNA sequences overlap with the targeted TF binding sites to be effective. 2) There are multiple GATA1 and TAL1 binding sites within the targeted HS2 enhancer, thus the targeting of dCas9 by a single sgRNA may not be sufficient to block all TF binding activities.

We observed that the targeting of dCas9 repressors (e.g. dCas9-KRAB, dCas9-LSD1 and enCRISPRi) to HS2 enhancer impaired GATA1 and TAL1 binding by ChIP-seq (Fig. 3 and Supplementary Fig. 3). Importantly, dCas9 repressors induced profound epigenetic alterations (e.g. loss of H3K4me1/2 and H3K27ac, and/or gained H3K9me3) beyond the targeted HS2 enhancer at the β -globin gene cluster (Fig. 3 and Supplementary Fig. 2). Based on these findings, we postulated that the impaired GATA1/TAL1 binding is likely due to altered epigenetic landscape instead of direct competition with dCas9 binding. We have included the discussion of these points in the revised manuscript (page 8).

- For enCRISPRi the authors produced and tested both possible combinations of the dual effectors (indicated as LK and KL). Would be interesting to discuss what would happen with the positive effectors if more than one was tested together.

We appreciate this thought, and have performed similar analyses of the enCRISPRa system by comparing individual dCas9-effectors (dCas9-VP64 and dCas9-p300) with dual-effector dCas9-p300 + MCP-VP64 (PV) and dCas9-VP64 + MCP-p300 (VP) (Fig. 1a-c). Specifically, at the *MYOD* enhancer, while dCas9-VP64 (V) had no effect on *MYOD* expression, dCas9-p300 (P) significantly upregulated *MYOD* compared to dCas9 alone or the non-transduced control cells (Fig. 1b). More importantly, the dual-effector enCRISPRa systems dCas9-p300 + MCP-VP64 (PV) and dCas9-VP64 + MCP-p300 (VP) outperformed individual effectors (dCas9-VP64 and dCas9-p300), resulting in more pronounced upregulation of *MYOD* expression (Fig. 1b). Similar results were observed at an independent enhancer (HS2 enhancer at the β -globin LCR), in which the dual-effector enCRISPRa systems led to more significant upregulation of the linked β -like globin genes *HBE1*, *HBG1/2* and *HBB* than the individual effectors (Fig. 1c). These new results demonstrate that the combinations of dual effectors (VP64 and p300) are also associated with more potent transcriptional perturbations at the targeted *MYOD* and β -globin enhancers.

-In the statement “but also identify selective ‘vulnerabilities’ of enhancers that may be employed to precisely control gene expression”. What are they kind of “vulnerabilities” the authors meant?

We have elaborated this statement with more details. We postulate that the functionally relevant constituent enhancers within super-enhancer clusters and/or the specific TF binding sites within constituent enhancers may be selective ‘vulnerabilities’ for perturbation of enhancer function.

-You mention therapeutics briefly in discussion but while possibly OK for enCRISPRi, might be too high levels for some things in terms of enCRISPRa and can be dangerous for the patient. Also, don't think these constructs could fit into the commonly used therapeutic AAV vector. All of this should be mentioned and discussed.

We appreciate these excellent thoughts, and have added a new paragraph to discuss these points in the Discussion under “Considerations for Enhancer Perturbations by CRISPR Epigenetic Editing” (pages 14-15).

-Figure 1B and C – lacking in figure or description n=?

We provided the description of *N* in Fig. 1 and all other relevant figures.

-Figure 1C – distance scales will be helpful

We provided distance scales as suggested.

-Figure 1D – aside from the on- and off-target loci, there are a couple of loci coming up with over a 100 counts. It would be interesting to elaborate on those, rather they are in the HBG locus or outside of it, and rather those are contact point to other genes. This information is crucial for later in vivo tests.

We appreciate this thoughtful comment, and have performed additional experiments to address this important question (Supplementary Fig. 1e-h). More specifically, as the reviewer noted, there are several loci with over 100 read counts in dCas9-p300 ChIP-seq in 293T cells (enCRISPRa from Fig. 1f) using target-specific sgRNA (sgHS2) or non-targeting sgGal4. The chromosome coordinates of the top 5 loci are labelled, which are located in the proximity of 4 genes (*EP300*, *ANKRD20A2*, *ANKRD30BL* and *ZNF806*). These genomic loci or genes do not appear to be related to the targeted β -globin genes or enhancers, thus are likely due to off-target dCas9 binding. Importantly, we observed no significant change in expression of the off-target-associated genes by qRT-PCR, suggesting that the dCas9 off-target binding did not significantly impact the transcription of nearest neighbor genes (Supplementary Fig. 1e,f). We performed similar analyses in K562 cells expressing enCRISPRi, and observed no significant change in gene expression of potential dCas9-KRAB off-target binding (Supplementary Fig. 1g,h). Based on these findings, we included a paragraph to discuss important considerations for enhancer perturbation using enCRISPRi/a including the assessment of potential off-target effects (pages 14-15).

-Figure 2 – The authors claim that HBB and HBG1 are affected by the enCRISPRi. If so, then it seems that many other genes are affected as well. Maybe another representation of the data is better where both log2FC and the pValue is plotted?

As noted by the reviewer, although the expression of *HBE1*, *HBG1/2* and *HBB* genes were affected by enCRISPRi, there were other genes showing variations in gene expression in sgHS2 vs sgGal4 expressing cells, especially the genes with relatively low RNA-seq signals (Fig. 2c). We reasoned that some of the observed expression changes may be due to technical variations of RNA-seq or dCas9 off-target effects (as discussed above). Nonetheless, the side-by-side comparisons show more pronounced effects on the expression of β -globin genes upon enCRISPRi compared to dCas9-KRAB or dCas9-LSD1 alone. We opted to use plotting with RNA-seq signal (x-axis) vs log2FC (y-axis) because one can appreciate the gene expression changes as related to expression levels, which cannot be represented in standard volcano plots (e.g. using log2FC vs p-value). Similar plots were also used in previous publications for similar purposes^{2,3}. We also included a paragraph in the revised manuscript to discuss important considerations for enhancer perturbation using enCRISPRi/a including the assessment of potential off-target effects

-Figure 2E - please check the scales of the tracks indicated in the zoomed and expanded images. For example the DHS in the zoomed is 0-300 while in the expanded track is 0-80 and in both it seems to reach the maximum but doesn't exceed it.

We have checked and confirmed that all the scales were used appropriately in Fig. 2e. We used smaller scales for the zoom-out view to better display the relatively weaker signals at some *cis*-elements (e.g. 0-80 for DHS). The peak summits of some regions may be difficult to visualize due to image compression. The raw (fastq) and processed (bigwig) files of all the genomic datasets are deposited in GEO (GSE132216) and listed in Supplementary Table 1.

-In Figure 3 enCRISPRi(LK) – (C/K/L/KL) tracks. Please write a description either in the figure legend or in the Methods

We provided the description in the figure legends.

-Figure 5D – use either Spi1/PU.1 or PU.1 in text and figure at all places

We use Spi1 in all the text and figures, except when first introducing the gene “Spi1 (or PU.1)” in page 11.

Reviewer #3 (Remarks to the Author):

In this manuscript, Li et al. describe the development and testing of enhanced systems for CRISPRi and CRISPRa with the goal of modulating enhancer activity. The method combines dCas9 tagging with the MS2-MCP system for recruiting additional regulatory domains. They show efficacy at a handful of enhancers *in vitro* and perform a small screen *in vivo*. The authors also analyze the effects on histone modifications and TF binding at the targeted loci. While the study is overall well designed, we don't believe that it warrants publication in Nature Communications as this is a very small advance in the context of CRISPR-based epigenome editing.

We understand that this reviewer's main concern relates to the novelty of the methods. Here we highlight several conceptual and technical advances that this study yields in light of recent publications:

- (a) While the strategies (dCas9 fusing and MS2-MCP scaffolding) and individual effectors (VP64, p300, KRAB and LSD1) have been used in other CRISPR-based epigenetic editing systems (see recent reviews by Gersbach⁴ and Qi⁵), the combinations to use enhancer-targeting epigenetic enzymes and transcriptional effector domains (e.g. p300+VP64 in enCRISPRa and LSD1+KRAB in enCRISPRi) have not been tested in the context of enhancer activation/repression. More importantly, previous findings were based mainly on *in vitro* cell models with limited insights into the efficacy and applicability of dCas9 epigenetic editing during *in vivo* development or disease process. These new aspects of dCas9-based epigenetic editing are the major focuses of this study.
- (b) A key question that was not addressed in previous reports is whether the combinations of enhancer-targeting epigenetic enzymes (e.g. LSD1 and p300) and other effector domains may enhance the efficacy through epigenetic mechanisms. In this study, we have systematically examined the changes in epigenetic landscapes by performing 90 individual CHIP-seq experiments (Supplementary Table 1). Our results demonstrate that independent repressors (LSD1 and KRAB) act through distinct mechanisms to modulate gene transcription by re-writing epigenetic landscapes at the targeted genomic loci. The combined effects on H3K9me3 deposition and H3K4me1/2 removal exceeded individual effectors (Fig. 3; Supplementary Figs. 2 and 5), illustrating that

different repressor domains act through independent mechanisms for maximal enhancer perturbation. Hence, our results reveal new insights into the cooperative activities between enhancer-targeting epigenetic modifiers and other effector domains by impacting epigenetic landscapes at the targeted genomic loci.

- (c) Moreover, by allele-specific activation or repression of an oncogenic super-enhancer in human T-ALL cells, our new results demonstrate the efficacy of enCRISPRa and enCRISPRi for functional analysis of disease-associated enhancers (Fig. 4). These studies expanded previous analyses and provided the proof-of-principle for further development of dCas9-based epigenetic editing tools to control disease phenotypes by targeting non-coding regulatory elements.
- (d) Most importantly, we generated a new dCas9-KRAB knock-in (KI) mouse model, and demonstrated the *in vivo* efficacy of enCRISPRi for single-locus and multi-loci enhancer perturbation screens. Our results not only identify previously unknown lineage-specific enhancers required for hematopoiesis, but also provide a new enCRISPRi mouse model for functional prioritization of CREs which can be extended to other cell types or human diseases. To our knowledge, these studies represent the first set of functional analysis of developmental enhancers in the differentiation of hematopoietic stem cells by dCas9-based epigenetic editing *in vivo*.

In our view, these findings and others included in this manuscript established a set of improved dCas9-based epigenetic editing tools for targeted enhancer modulation in development and disease. Given that we have tested the efficacy of these systems in multiple cell models *in vitro*, in xenografts and *in vivo*, we believe that the strategies and concepts developed in this study should be broadly applicable to other model systems, as commented by Reviewer 1: “...the synergistic system and the development of a mouse model significantly adds to the CRISPR toolbox and is expected to be adapted by many investigators beyond the specialist field of gene regulation.”

1. The authors describe a “new” CRISPR-based epigenome engineering approach that they term enCRISPRi and enCRISPRa, which involves dCas9 fused to regulatory domains with additional MCP fused regulatory domains recruited via MS2 loops on guide RNAs. This strategy was published by Konermann et al in 2015 and named SAM. While the authors have changed the regulatory domains (to other previously published domains), the concept is exactly the same and therefore represents a minor advance. In addition, there are several papers that describe the use of multiple regulatory domains at the same time (Konermann et al Nature 2015, Chavez et al Nature Methods 2015, Carleton et al Cell Systems 2017, and Yeo et al Nature Methods 2018) and multiplex regulation, the regulation of multiple genes simultaneously, has been shown by at least three groups (Konermann et al Nature 2015, Carleton et al Cell Systems 2017, and Yeo et al Nature Methods 2018). While the *in vivo* section adds some novelty, the experiment is more similar to an *ex vivo* study (where the epigenetic engineering is occurring) with *in vivo* phenotyping.

We agree with the reviewer that overall concept on engineering dual-effector-based dCas9 complex using MS2-MCP scaffolding is similar to Konermann’s study⁶, and we have significantly revised the texts to discuss existing concepts and tools of CRISPR-based epigenetic editing. It is important to note that there are key differences between our studies and published work in both the technical details and conceptual findings, as detailed below. We believe that the new strategies of combining enhancer-targeting epigenetic enzymes and other effector domains (p300+VP64 and LSD1+KRAB) are important additions to the existing dCas9 tools. Moreover, the in-depth analyses of enCRISPRi/a-mediated enhancer perturbations in multiple models *in vitro*, in xenografts and *in vivo* extended the scope of previous studies to more physiologically relevant contexts. More specifically:

- (a) Differences from the SAM system: the SAM system employs dCas9-VP64 fusion protein together with MCP-p65-HSF1 to activate gene expression⁶. No epigenetic regulators and no repressive complexes were described, which are the main focuses of this study.
- (b) Differences from other published studies using multiple regulatory domains: various regulatory domains have been tested individually or in combinations, including dCas9-VPR⁷, SAM⁶, dCas9-p300⁸, dCas9-VP160⁹, dCas9-KRAB-MeCP2¹⁰ and SID-dCas9-KRAB³; however, no study has tested the combination of enhancer-targeting epigenetic enzymes (p300 and LSD1) and transcriptional effector domains (VP64 and KRAB) for enhancer activation and repression *in vitro* and *in vivo*. These are the main focuses of this study.
- (c) Re *in vivo* enhancer perturbation: there is an important mis-interpretation of our *in vivo* study. As depicted in Figs. 5d and 6a, the primary HSPCs were transduced with pooled sgRNAs followed by bone marrow transplantation (BMT) to recipient mice. Note that the expression of the dCas9-KRAB knock-in transgene was not activated *ex vivo*, instead it was activated by Dox administration for 16 weeks in the recipient mice *in vivo*. Together with CRE-targeting sgRNAs, the assembled enCRISPRi complex leads to epigenetic repression in differentiating HSCs *in vivo*. We specifically designed these experiments to determine the requirement of CRE

repression in HSC function *in vivo* without confounding factors from *ex vivo* culture and/or BMT (e.g. HSC homing or effects from short-lived progenitors).

2. Related to the point above, the manuscript is written in such a way that makes the developed method appear completely unique. The introduction and conclusion set the tone that these enCRISPR methods are special in their capabilities, but several studies have used variants of CRISPRa and CRISPRi successfully at enhancers (e.g. Hilton et al 2014, Klann et al, Carleton et al Cell Systems 2017).

As detailed in the responses to the general comments and point #1, we have significantly revised the texts including introduction, conclusion and discussion to better describe the existing literature/knowledge, and the technical and conceptual advances that our work presents in light of previous publications.

3. The term “enhanced crispr”, which is used in the title and name of the method, has already been used and should be changed to avoid confusion.

We noted that the term “enhanced CRISPR” have been used in the literature a few times when describing Cas9-mediated genome editing by HDR^{11, 12} or dCas9-based repression¹⁰, but did not refer to one specific technology. Nevertheless, we have changed the terminology to “enhancer-targeting CRISPR epigenetic editing” and kept the same acronyms “enCRISPRi” and “enCRISPRa” to avoid confusions.

4. Guide RNAs without MS2 loops should be used as a control. In addition, we could not find a direct comparison of enCRISPRa to P300 and VP64 alone.

We appreciate these suggestions, and have performed the following new experiments to address these comments:

- (a) sgRNAs without MS2 as controls: we compared the effects on target gene expression by targeting dCas9 alone (control) or enCRISPRa (dCas9-p300 + MCP-VP64) together with sgRNAs with or without MS2 loops to two independent enhancers *MYOD* and *HS2* (Supplementary Fig. 1c,d). We noted that sgRNAs with MS2 loops together with MCP-VP64, when combined with dCas9 alone or enCRISPRa, significantly enhanced target gene expression compared with sgRNAs without MS2. The combinations of enCRISPRa (dCas9-p300 + MCP-VP64) and sgRNAs with MS2 loops resulted in the most significant gene upregulation (Supplementary Fig. 1c,d). These results demonstrate that the MS2 loops are required for the recruitment of MCP-fused effectors to dCas9 complexes.
- (b) Direct comparisons of enCRISPRa to P300 and VP64 alone: as suggested, we compared individual dCas9-effectors (dCas9-VP64 and dCas9-p300) with dual-effector enCRISPRa systems including dCas9-p300 + MCP-VP64 (PV) and dCas9-VP64 + MCP-p300 (VP) (Fig. 1a-c). At the *MYOD* enhancer, while dCas9-VP64 (V) had no effect on *MYOD* expression, dCas9-p300 (P) significantly upregulated *MYOD* compared to dCas9 alone or the non-transduced control cells (Fig. 1b). More importantly, the dual-effector enCRISPRa systems dCas9-p300 + MCP-VP64 (PV) and dCas9-VP64 + MCP-p300 (VP) outperformed individual effectors (dCas9-VP64 and dCas9-p300), resulting in more pronounced upregulation of *MYOD* expression (Fig. 1b). Similar results were observed at the β -globin *HS2* enhancer, in which the dual-effector enCRISPRa systems led to more significant upregulation of the β -globin genes *HBE1*, *HBG1/2* and *HBB* than the individual effectors (Fig. 1c). These new results demonstrate that the combinations of dual effectors (VP64 and p300) are associated with more potent transcriptional perturbations at the targeted *MYOD* and β -globin enhancers.

5. In the introduction, the authors state that the effect of dCas9 fusions declines rapidly when the target regions move away from the proximal promoter. But references 18-20 don't really show this.

In the original reference #19 (Gilbert et al. 2014 Cell)¹³, by tiling sgRNA screen using CRISPRi (dCas9-KRAB; Fig. 1C) or CRISPRa (dCas9-VP64; Fig. 3B), the authors showed that sgRNA position relative to TSS is a critical factor in determining efficacy of CRISPRi/a phenotypes, respectively. As the sgRNA moves away from TSS, the phenotype relative to control rapidly declines. In reference #18 (Gilbert et al., 2013 Cell)¹⁴, the authors used a few sgRNAs at varying distance to the TSS of *CXCR4* gene or an EGFP reporter, and noted a trend of decreased CRISPRi activity as the sgRNA locates more distal to TSS (Figs. 3B and 4A). Similarly in reference #20 (Zalatan et al., 2015 Cell)¹⁵, the authors noted decreased CRISPRa activity as the sgRNAs locates more distal to TSS (Fig. S3B). However, no statistical analyses were performed and the results on sgRNA position relative to TSS were not discussed in these studies^{14, 15}. In addition, in reference #21 (Koneremann et al., 2015 Nature)⁶, the authors showed that the efficacy of SAM-mediated gene activation rapidly declines as the sgRNA position locates further away from TSS at multiple independent genes or lincRNA loci (Fig. 2a,d and Extended Data Fig. 3). Based on these findings, we have updated the citations to only include references #19 and #21 to avoid confusions.

6. The authors state that the same guide RNAs were used in each experiment shown in Figure 1. Do these guide RNAs all include MS2 loops?

We apologize for the confusion. The same sgRNAs with MS2 loops were used in experiments comparing dual-effector enCRISPRa systems and individual effectors (dCas9-p300 and dCas9-VP64) (Figs. 1b,c), or dCas9 ChIP-seq experiments (Fig. 1f). In experiments comparing different dCas9 activators, sgRNAs with MS2 loops were used for enCRISPRa and SAM, whereas the same sgRNAs without MS2 were used for other systems (dxCas9-VPR and SunTag) (Figs. 1d,e). As detailed in the responses to point #4, we have performed new experiments comparing sgRNAs with and without MS2 (Supplementary Fig. 1c,d). The results demonstrate that the MS2 loops are required for the recruitment of MCP-fused effectors to the dCas9 complexes, but the presence of MS2 loops alone (e.g. without MCP-fused effectors) have no significant effect on dCas9-effector-mediated perturbations. We have revised the texts and legends to be more specific.

7. The authors state that H3K27me3 was not observed at the beta-globin locus with or without CRISPRi techniques, but the data is not shown. The data should be shown or the statement should be left out.

As suggested, we included the H3K27me3 ChIP-seq results in Fig. 3 and Supplementary Figs. 2 and 4. Of note, no enrichment of H3K27me3 was observed at the β -globin gene cluster with or without various dCas9 repressors, whereas significant enrichment of H3K27me3 was observed at other loci including *LINC01039* and *ATP8B1*, indicating the validity of the ChIP-seq datasets. All the raw and processed ChIP-seq were uploaded to GEO (GSE132216) and listed in Supplementary Table 1.

8. In several places in the manuscript (bottom of page 7, 2nd paragraph of page 8, and 1st paragraph of the discussion), the authors state that there is cooperation between repressor domains, but this hasn't been shown. The additional repression observed using enCRISPRi could be due to multiple copies of the MCP fused receptor and not cooperation or synergy. The authors should use an unfused dCas9 with MCP fusions as a comparison.

Several lines of evidence suggest that the combinations of different repressor domains may act cooperatively to repress enhancers, such as: (1) we initially noted the combinations of dCas9-LSD1 + MCP-KRAB (enCRISPRi-LK) or dCas9-KRAB + MCP-LSD1 (enCRISPRi-KL) led to more significant downregulation of target genes compared to individual effectors (dCas9-KRAB and dCas9-LSD1) (Fig. 2b,c). (2) By ChIP-seq analyses, we noted that dCas9-KRAB resulted in increases in H3K9me3 at the targeted HS2 enhancer, whereas dCas9-LSD1 had no effect on H3K9me3 deposition but instead modestly decreased H3K4me1/2. More importantly, enCRISPRi (LK and KL) resulted in broader enrichment of H3K9me3 and corresponding loss of active histone marks (H3K4me1/2 and H3K27ac) (Fig. 3 and Supplementary Fig. 2). Nonetheless, we agree with the reviewer that these results did not provide strong evidence to suggest cooperativity or synergy between repressor domains, and we have revised the relevant texts to avoid confusions.

In addition, we have included unfused dCas9 with MCP fusions together with sgRNAs without MS2 loops as additional controls as suggested (Supplementary Fig. 1c,d).

9. The authors state that enCRISPRi effects are constrained by CTCF, since CTCF binding is not affected. However, the authors do not have any evidence that CTCF has anything to do with confinement. The authors should simply state that CTCF binding is unaffected by enCRISPRi.

We appreciate this thoughtful comment, and have revised the statement as suggested.

10. ChIP-seq signal at control regions that were not targeted should be shown (possibly as a supplemental figure) to assure comparable signal between ChIP-seq data sets overall.

As suggested, we included the genome browser views of all the ChIP-seq datasets in non-targeted control regions including *LINC01039* and *ATP8B1* in Supplementary Fig. 4. Comparable ChIP-seq signals between control and different dCas9-effector expressing cells were observed at the non-targeted regions.

11. In the section on allele-specific activation of TAL1, the results look promising but allele-specificity hasn't been proven. Allele-specific targeting should be shown by ChIP and a control of a wildtype cell not responding to the mutant guide RNAs should be shown.

We appreciate these thoughtful comments, and performed additional experiments to address this comment. Specifically, to validate the allele-specific binding of enCRISPRa by mutant (Mut) allele-specific sgRNAs (sgMut1 and sgMut2), we performed dCas9 ChIP experiments in Jurkat and K562 cells, respectively (Fig. 4b). In Jurkat cells that carry both WT and Mut alleles, expression of sgWT1 or sgWT2 with enCRISPRa led to significant and comparable dCas9 binding at both alleles relative to the non-targeting sgGal4. By contrast, sgMut1 or sgMut2

resulted in significant dCas9 binding at the Mut but not the WT allele (Fig. 4b). In K562 cells with only WT alleles, expression of sgWT1 or sgWT2 led to significant dCas9 binding at the WT allele, whereas no significant dCas9 binding at either allele was noted with the expression of sgMut1 or sgMut2. These results provide direct evidence that sgMut1 or sgMut2 targets the dCas9 complexes specifically to the Mut allele, while sgWT1 or sgWT2 targets dCas9 to both WT and Mut alleles.

12. In Figure 4E, why is there cell luminescence observed at 4 hours post implantation? If there is luminescence 4 hours post implantation, why is there no luminescence at 0 weeks post-transplant in Figure 4F?

We have provided more technical details of the bioluminescence experiments in the Methods (page 21). Briefly, the measurement at 4 hour post-xenograft was to confirm the successful xenotransplantation of the luciferase-expressing leukemia cells. Since most of the xenografted leukemia cells will not survive during the first few days, the bioluminescence signal usually drops to nearly undetectable levels by the next measurement at 72~96 hour (week 0 in Fig. 4i,j). As the surviving leukemia cells proliferate over time, the bioluminescence signals increase in subsequent measurements (weeks 2 and 4; Fig. 4i,j). This method has been widely used to examine the activity of human leukemia cells *in vivo*¹⁶.

13. In the screens in the last two sections of the results, the authors often interpret no effect on phenotype as the enhancer not playing a functional role. It is very difficult to interpret negative results from this type of experiment. If the authors want to conclude negative results (e.g. that E1 and E3 of *Cebpa* are dispensable), then efficient targeting of these sites needs to be shown to rule out the simple explanation that the guide RNAs targeting the dispensable sites simply do not work.

We appreciate this insightful comment. We agree with the reviewer that the lack of gene repression and sgRNA depletion by enCRISPRi *in vivo* could be due to: 1) the sgRNAs did not efficiently target enCRISPRi to the enhancer sequences; and/or 2) the targeted enhancers were not essential for target gene expression. To distinguish between these possibilities, we determined the targeting of dCas9 complexes by ChIP experiments in primary HSPCs isolated from the dCas9-KRAB knock-in mouse bone marrow. Upon retroviral expression of each target-specific sgRNA or non-targeting sgGal4 in dCas9-KRAB expressing HSPCs, ChIP experiments were performed to determine the enrichment of dCas9 binding at the sgRNA-targeted genomic loci (Supplementary Fig. 11a-d). We found that expression of CRE-targeting sgRNAs resulted in significant increases in dCas9 binding at the targeted promoter or enhancer regions relative to non-targeting sgGal4. Importantly, the dCas9 targeting efficiency (as determined by the ChIP signals) was largely comparable for the tested sgRNAs at each locus, despite that only some sgRNAs resulted in significant downregulation of target genes (Supplementary Fig. 10). For example, the sgRNAs targeting *Cebpa* promoter (P-sg1) resulted in comparable dCas9 binding as the sgRNAs targeting *Cebpa* enhancers (E1-sg1, E1-sg2, E3-sg1 and E3-sg2; Supplementary Fig. 11a); however, only P-sg1 but not E1 or E3-targeting sgRNAs led to significant *Cebpa* downregulation (Supplementary Fig. 10a). Similar results were obtained for other tested loci including *Gata1*, *Gata2* and *Runx1* (Supplementary Fig. 11b-d). These findings suggest that the lack of target gene repression is unlikely due to inefficient dCas9/sgRNA targeting. It is important to note that the presence of dCas9 chromatin occupancy may not reflect the repressive function of the assembled enCRISPRi complexes on chromatin, and we have discussed this possibility in the revised manuscript.

14. As discussed in response 1 above, the authors state that they are performing multiplex perturbations; however, the field uses the word multiplex as the simultaneous (i.e. in the same cells) perturbation of multiple genes. This type of experiment has not been performed in this study, instead I would refer to the section on multiplex perturbations as a small screen to avoid confusion.

As suggested, we have changed the term "multiplex" to "pooled sgRNA-based screening".

15. Are the p-values in Figure S6 adjusted for multiple hypothesis testing? If not, the p-values should be corrected.

In the revised manuscript, we used false discovery rate (FDR) for *P*-value adjustments by multiple hypothesis testing. The FDR values were calculated by MAGECK test¹⁷ with the parameters of --norm-method total --gene-lfc-method mean.

16. In the first paragraph of the discussion, the authors state that epigenetic writer proteins specifically modulate histone modifications. However, VP16 and KRAB are fairly non-specific chromatin modifiers that recruit several different cofactors.

We apologize for the misunderstanding. We revised the sentence to be more specific that the epigenetic writer proteins refer to p300 and LSD1.

17. In the data shown in Figures 1-3, what is the timing of the experiment, with respect to Dox induction, for assaying expression or protein:DNA interactions?

We provided the technical details in the figure legends and/or Methods. Briefly, in the experiments (e.g. qRT-PCR, RNA-seq and ChIP) with Dox-inducible enCRISPRa or enCRISPRi expression, the cells were treated with 1 µg/mL of Dox for 72 hours to activate dCas9 expression and processed for downstream analyses.

18. In all bar graphs, every point should be shown as well (as was done in Figure 4B).

We appreciate this comment. Per the journal guidelines, we have revised all bar graphs to show individual data points if $N \leq 10$. All the source data underlying the bar graphs are included in the Source Data file.

References:

1. Shariati, S.A. *et al.* Reversible Disruption of Specific Transcription Factor-DNA Interactions Using CRISPR/Cas9. *Molecular cell* **74**, 622-633.e624 (2019).
2. Thakore, P.I. *et al.* RNA-guided transcriptional silencing in vivo with *S. aureus* CRISPR-Cas9 repressors. *Nature communications* **9**, 1674 (2018).
3. Carleton, J.B., Berrett, K.C. & Gertz, J. Multiplex Enhancer Interference Reveals Collaborative Control of Gene Regulation by Estrogen Receptor alpha-Bound Enhancers. *Cell systems* **5**, 333-344.e335 (2017).
4. Pickar-Oliver, A. & Gersbach, C.A. The next generation of CRISPR-Cas technologies and applications. *Nature reviews. Molecular cell biology* **20**, 490-507 (2019).
5. Xu, X. & Qi, L.S. A CRISPR-dCas Toolbox for Genetic Engineering and Synthetic Biology. *Journal of molecular biology* **431**, 34-47 (2019).
6. Konermann, S. *et al.* Genome-scale transcriptional activation by an engineered CRISPR-Cas9 complex. *Nature* **517**, 583-588 (2015).
7. Chavez, A. *et al.* Highly efficient Cas9-mediated transcriptional programming. *Nature reviews. Molecular cell biology* **12**, 326-328 (2015).
8. Hilton, I.B. *et al.* Epigenome editing by a CRISPR-Cas9-based acetyltransferase activates genes from promoters and enhancers. *Nature biotechnology* **33**, 510-517 (2015).
9. Chavez, A. *et al.* Comparison of Cas9 activators in multiple species. *Nature methods* **13**, 563-567 (2016).
10. Yeo, N.C. *et al.* An enhanced CRISPR repressor for targeted mammalian gene regulation. *Nature methods* **15**, 611-616 (2018).
11. Liang, X., Potter, J., Kumar, S., Ravinder, N. & Chesnut, J.D. Enhanced CRISPR/Cas9-mediated precise genome editing by improved design and delivery of gRNA, Cas9 nuclease, and donor DNA. *Journal of biotechnology* **241**, 136-146 (2017).
12. Farboud, B. *et al.* Enhanced Genome Editing with Cas9 Ribonucleoprotein in Diverse Cells and Organisms. *Journal of visualized experiments : JoVE* (2018).
13. Gilbert, L.A. *et al.* Genome-Scale CRISPR-Mediated Control of Gene Repression and Activation. *Cell* **159**, 647-661 (2014).
14. Gilbert, L.A. *et al.* CRISPR-mediated modular RNA-guided regulation of transcription in eukaryotes. *Cell* **154**, 442-451 (2013).
15. Zalatan, J.G. *et al.* Engineering complex synthetic transcriptional programs with CRISPR RNA scaffolds. *Cell* **160**, 339-350 (2015).
16. Deng, M. *et al.* LILRB4 signalling in leukaemia cells mediates T cell suppression and tumour infiltration. *Nature* **562**, 605-609 (2018).
17. Li, W. *et al.* MAGECK enables robust identification of essential genes from genome-scale CRISPR/Cas9 knockout screens. *Genome biology* **15**, 554 (2014).

Reviewers' Comments:

Reviewer #1:

Remarks to the Author:

the authors have sufficiently addressed my previous comments. I recommend publication.

Reviewer #2:

Remarks to the Author:

The authors have adequately addressed all my comments.

Reviewer #3:

Remarks to the Author:

The revised manuscript is improved from the last version. While our technical concerns about this manuscript have been diminished, we still feel that the novelty is lacking. The authors mention being the first to use a combination of epigenetic enzymes in combination with "transcriptional effectors", this seems like a very minor distinction. While we agree that the mouse experiment is mostly done in vivo, the guide RNA delivery is performed ex vivo, which means that the method is only applicable to very specialized "in vivo" experiments and does not generally lead to "efficient analysis of enhancer function...in vivo" as mentioned in the abstract. It appears that we might be in the minority on the issue of novelty and ultimately leave that decision up to the editor.

Specific comments:

In the third paragraph of the introduction, the authors write, "While the first-generation dCas9 activator or repressor complexes such as dCas9-VP64 and dCas9- KRAB can effectively modulate transcription when tethered to gene-proximal promoters, the effect declines rapidly when its target region moves away from proximal promoter sequences 19, 21. This is likely because VP64 or KRAB preferentially interferes with the basal transcription initiation and/or elongation apparatus operating at gene promoters 29, 31. Since distal CREs such as enhancers may not rely on the basal transcription apparatus, these methods were ineffective and variable in modulating enhancer activity."

This is a very misleading interpretation of existing studies and should be heavily edited or removed. While activity at promoters decreases as the targeting moves away from the transcription start site, this is irrelevant to the distance from transcription start sites that enhancers typically reside. Enhancers do recruit general transcription machinery and likely require it. In addition, many groups have used dCas9-KRAB and dCas9-VP64 successfully at enhancers.

On the bottom of page 15, CRISPR is misspelled CPRISR.

Here we provide our point-by-point response (in *blue*) to the reviewers' comments.

Reviewer #1 (Remarks to the Author):

The authors have sufficiently addressed my previous comments. I recommend publication.

Reviewer #2 (Remarks to the Author):

The authors have adequately addressed all my comments.

Reviewer #3 (Remarks to the Author):

The revised manuscript is improved from the last version. While our technical concerns about this manuscript have been diminished, we still feel that the novelty is lacking. The authors mention being the first to use a combination of epigenetic enzymes in combination with “transcriptional effectors”, this seems like a very minor distinction. While we agree that the mouse experiment is mostly done *in vivo*, the guide RNA delivery is performed *ex vivo*, which means that the method is only applicable to very specialized “*in vivo*” experiments and does not generally lead to “efficient analysis of enhancer function...*in vivo*” as mentioned in the abstract. It appears that we might be in the minority on the issue of novelty and ultimately leave that decision up to the editor.

Specific comments:

In the third paragraph of the introduction, the authors write, “While the first-generation dCas9 activator or repressor complexes such as dCas9-VP64 and dCas9- KRAB can effectively modulate transcription when tethered to gene-proximal promoters, the 4 effect declines rapidly when its target region moves away from proximal promoter sequences 19, 21. This is likely because VP64 or KRAB preferentially interferes with the basal transcription initiation and/or elongation apparatus operating at gene promoters 29, 31. Since distal CREs such as enhancers may not rely on the basal transcription apparatus, these methods were ineffective and variable in modulating enhancer activity.”

This is a very misleading interpretation of existing studies and should be heavily edited or removed. While activity at promoters decreases as the targeting moves away from the transcription start site, this is irrelevant to the distance from transcription start sites that enhancers typically reside. Enhancers do recruit general transcription machinery and likely require it. In addition, many groups have used dCas9-KRAB and dCas9-VP64 successfully at enhancers.

We removed these sentences to avoid misinterpretation, as suggested.

On the bottom of page 15, CRISPR is misspelled CPRISR.

We corrected the typo.